# Learning Disentangled Representation by Exploiting Pretrained Generative Models: A Contrastive Learning View

**Xuanchi Ren**[1*], **Yang Tao**[2*], **Yuwang Wang**[3†], **Wenjun Zeng**[4]
[1]HKUST, [2]Xi'an Jiaotong University, [3]Microsoft Research Asia, [4]EIT
[1] `xrenaa@connect.ust.hk`  [2] `yt14212@stu.xjtu.edu.cn`
[3] `yuwwan@microsoft.com`  [4] `wenjunzeng@eias.ac.cn`

## Abstract

From the intuitive notion of disentanglement, the image variations corresponding to different factors should be distinct from each other, and the disentangled representation should reflect those variations with separate dimensions. To discover the factors and learn disentangled representation, previous methods typically leverage an extra regularization term when learning to generate realistic images. However, the term usually results in a trade-off between disentanglement and generation quality. For the generative models pretrained without any disentanglement term, the generated images show semantically meaningful *variations* when traversing along different directions in the latent space. Based on this observation, we argue that it is possible to mitigate the trade-off by $(i)$ leveraging the pretrained generative models with high generation quality, $(ii)$ focusing on discovering the traversal directions as factors for disentangled representation learning. To achieve this, we propose **Dis**entaglement via **Co**ntrast (`DisCo`) as a framework to model the variations based on the target disentangled representations, and contrast the variations to jointly discover disentangled directions and learn disentangled representations. `DisCo` achieves the state-of-the-art disentangled representation learning and distinct direction discovering, given pretrained non-disentangled generative models including GAN, VAE, and Flow. Source code is at `https://github.com/xrenaa/DisCo`.

## 1 Introduction

Disentangled representation learning aims to identify and decompose the underlying explanatory factors hidden in the observed data, which is believed by many to be the only way to understand the world for AI fundamentally (Bengio & LeCun, 2007). To achieve the goal, as shown in Figure 1 (a), we need an encoder and a generator. The encoder to extract representations from images with each dimension corresponds to one factor individually. The generator (decoder) decodes the changing of each factor into different kinds of image variations.

With supervision, we can constrain each dimension of the representation only sensitive to one kind of image variation caused by changing one factor respectively. However, this kind of exhaustive supervision is often not available in real-world data. The typical unsupervised methods are based on a generative model to build the above encoder and generator framework, e.g., VAE (Kingma & Welling, 2014) provides encoder and generator, and GAN (Goodfellow et al., 2014; Miyato et al., 2018; Karras et al., 2019) provides generator. During the training process of the encoder and generator, to achieve disentangled representation, the typical methods rely on an additional disentanglement regularization term, e.g., the total correlation for VAE-based methods (Higgins et al., 2017; Burgess et al., 2018; Kumar et al., 2017; Kim & Mnih, 2018; Chen et al., 2018) or mutual information for InfoGAN-based methods (Chen et al., 2016; Lin et al., 2020).

---

[*]Equal contribution. Work done during internships at Microsoft Research Asia.
[†]Corresponding author

Figure 1: (a) The encoder and generator framework for learning disentangled representation. (b) Our alternative route to learn disentangle representation with fixed generator.

However, the extra terms usually result in a trade-off between disentanglement and generation quality (Burgess et al., 2018; Khrulkov et al., 2021). Furthermore, those unsupervised methods have been proved to have an infinite number of entangled solutions without introducing inductive bias (Locatello et al., 2019). Recent works (Shen & Zhou, 2021; Khrulkov et al., 2021; Karras et al., 2019; Härkönen et al., 2020; Voynov & Babenko, 2020) show that, for GANs purely trained for image generation, traversing along different directions in the latent space causes different variations of the generated image. This phenomenon indicates that there is some disentanglement property embedded in the latent space of the pretrained GAN. The above observations indicate that training the encoder and generator simultaneous may not be the best choice.

We provide an alternative route to learn disentangled representation: fix the pretrained generator, jointly discover the factors in the latent space of the generator and train the encoder to extract disentangled representation, as shown in Figure 1(b). From the intuitive notion of disentangled representation, similar image variations should be caused by changing the same factor, and different image variations should be caused by changing different factors. This provide a novel contrastive learning view for disentangled representation learning and inspires us to propose a framework: **Dis**entanglement via **Co**ntrast (`DisCo`) for disentangled representation learning.

In `DisCo`, changing a factor is implemented by traversing one discovered direction in the latent space. For discovering the factors, `DisCo` adopts a typical network module, *Navigator*, to provides candidate traversal directions in the latent space (Voynov & Babenko, 2020; Jahanian et al., 2020; Shen et al., 2020). For disentangled representation learning, to model the various image variations, we propose a novel $\Delta$-*Contrastor* to build a *Variation Space* where we apply the contrastive loss. In addition to the above architecture innovations, we propose two key techniques for `DisCo`: $(i)$ an entropy-based domination loss to encourage the encoded representations to be more disentangled, $(ii)$ a hard negatives flipping strategy for better optimization of Contrastive Loss.

We evaluate `DisCo` on three major generative models (GAN, VAE, and Flow) on three popular disentanglement datasets. `DisCo` achieves the state-of-the-art (SOTA) disentanglement performance compared to all the previous discovering-based methods and typical (VAE/InfoGAN-based) methods. Furthermore, we evaluate `DisCo` on the real-world dataset FFHQ (Karras et al., 2019) to demonstrate that it can discover SOTA disentangled directions in the latent space of pretrained generative models.

Our main contributions can be summarized as: $(i)$ To our best knowledge, `DisCo` is the first unified framework for jointly learning disentangled representation and discovering the latent space of pretrained generative models by contrasting the image variations. $(ii)$ We propose a novel $\Delta$-Contrastor to model image variations based on the disentangled representations for utilizing Contrastive Learning. $(iii)$ DisCo is an unsupervised and model-agnostic method that endows non-disentangled VAE, GAN, or Flow models with the SOTA disentangled representation learning and latent space discovering. $(iv)$ We propose two key techniques for `DisCo`: an entropy-based domination loss and a hard negatives flipping strategy.

## 2 RELATED WORK

**Typical unsupervised disentanglement.** There have been a lot of studies on unsupervised disentangled representation learning based on VAE (Higgins et al., 2017; Burgess et al., 2018; Kumar et al., 2017; Kim & Mnih, 2018; Chen et al., 2018) or InfoGAN (Chen et al., 2016; Lin et al., 2020). These methods achieve disentanglement via an extra regularization, which often sacrifices the generation quality (Burgess et al., 2018; Khrulkov et al., 2021). VAE-based methods disentangle the variations by factorizing aggregated posterior, and InfoGAN-based methods maximize the mutual

information between latent factors and related observations. VAE-based methods achieve relatively good disentanglement performance but have low-quality generation. InfoGAN-based methods have a relatively high quality of generation but poor disentanglement performance. Our method supplements generative models pretrained without disentanglement regularization term with contrastive learning in the *Variation Space* to achieve both high-fidelity image generation and SOTA disentanglement.

**Interpretable directions in the latent space.** Recently, researchers have been interested in discovering the interpretable directions in the latent space of generative models without supervision, especially for GAN (Goodfellow et al., 2014; Miyato et al., 2018; Karras et al., 2020). Based on the fact that the GAN latent space often possesses semantically meaningful directions (Radford et al., 2015; Shen et al., 2020; Jahanian et al., 2020), Voynov & Babenko (2020) propose a regression-based method to explore interpretable directions in the latent space of a pretrained GAN. The subsequent works focus on extracting the directions from a specific layer of GANs. Härkönen et al. (2020) search for important and meaningful directions by performing PCA in the style space of StyleGAN (Karras et al., 2019; 2020). Shen & Zhou (2021) propose to use the singular vectors of the first layer of a generator as the interpretable directions, and Khrulkov et al. (2021) extend this method to the intermediate layers by Jacobian matrix. All the above methods only discover the interpretable directions in the latent space, except for Khrulkov et al. (2021) which also learns disentangled representation of generated images by training an extra encoder in an extra stage. However, all these methods can not outperform the typical disentanglement methods. Our method is the first to jointly learn the disentangled representation and discover the directions in the latent spaces.

**Contrastive Learning.** Contrastive Learning gains popularity due to its effectiveness in representation learning (He et al., 2020; Grill et al., 2020; van den Oord et al., 2018; Hénaff, 2020; Li et al., 2020; Chen et al., 2020). Typically, contrastive approaches bring representations of different views of the same image (positive pairs) closer, and push representations of views from different images (negative pairs) apart using instance-level classification with Contrastive Loss. Recently, Contrastive Learning is extended to various tasks, such as image translation (Liu et al., 2021; Park et al., 2020) and controllable generation (Deng et al., 2020). In this work, we focus on the variations of representations and achieve SOTA disentanglement with Contrastive Learning in the *Variation Space*. Contrastive Learning is suitable for disentanglement due to: $(i)$ the actual number of disentangled directions is usually unknown, which is similar to Contrastive Learning for retrieval (Le-Khac et al., 2020), $(ii)$ it works in the representation space directly without any extra layers for classification or regression.

## 3 DISENTANGLEMENT VIA CONTRAST

### 3.1 OVERVIEW OF DisCo

From the contrastive view of the intuitive notion of disentangled representation learning, we propose a DisCo to leverage pretrained generative models to jointly discover the factors embedded as directions in the latent space of the generative models and learn to extract disentangled representation. The benefits of leveraging a pretrained generative model are two-fold: $(i)$ the pretrained models with high-quality image generation are readily available, which is important for reflecting detailed image variations and downstream tasks like controllable generation; $(ii)$ the factors are embedded in the pretrained model, severing as an inductive bias for unsupervised disentangled representation learning.

DisCo consists of a *Navigator* to provides candidate traversal directions in the latent space and a $\Delta$-*Contrastor* to extract the representation of image variations and build a *Variation Space* based on the target disentangled representations. More specifically, $\Delta$-*Contrastor* is composed of two shared-weight Disentangling Encoders. The variation between two images is modeled as the difference of their corresponding encoded representations extracted by the Disentangling Encoders.

In the *Variation Space*, by pulling together the variation samples resulted from traversing the same direction and pushing away the ones resulted from traversing different directions, the *Navigator* learns to discover disentangled directions as factors, and Disentangling Encoder learns to extract disentangled representations from images. Thus, traversing along the discovered directions causes distinct image variations, which causes separated dimensions of disentangled representations respond.

Different from VAE-based or InfoGAN-based methods, our disentangled representations and factors are in two separate spaces, which actually does not affect the applications. Similar to the typical

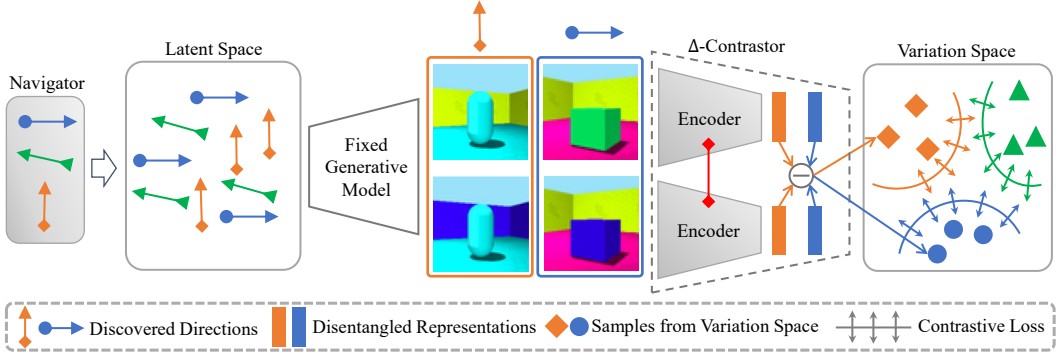

Figure 2: Overview of **DisCo**. DisCo consists of: ($i$) a *Navigator* exploring traversal directions in the latent space of a given pretrained generative model, ($ii$) a $\Delta$-*Contrastor* encoding traversed images into the *Variation Space*, where we utilize Contrastive Learning. Samples in the *Variation Space* correspond to the image variations along the directions provided by the *Navigator* labeled with different colors, respectively. $\Delta$-*Contrastor* includes two shared-weight Disentangling Encoders to extract disentangled representations respectively, and outputs the difference between the disentangled representations as variation representation. The Generative Model is fixed, and the *Navigator* and Disentangling Encoders marked with grey color are learnable.

methods, the Disentangling Encoder can extract disentangled representations from images, and the pretrained generative model with discovered factors can be applied to controllable generation. Moreover, DisCo can be applied to different types of generative models.

Here we provide a detailed **workflow** of DisCo. As Figure 2 shows, given a pretrained generative model $\boldsymbol{G}$: $\mathcal{Z} \rightarrow \mathcal{I}$, where $\mathcal{Z} \in \mathbb{R}^L$ denotes the latent space, and $\mathcal{I}$ denotes the image space, the workflow is: 1) A *Navigator* $\boldsymbol{A}$ provides a total of $D$ candidate traversal directions in the latent space $\mathcal{Z}$, e.g., in the linear case, $\boldsymbol{A} \in \mathbb{R}^{L \times D}$ is a learnable matrix, and each column is regarded as a candidate direction. 2) Image pairs $\boldsymbol{G}(\boldsymbol{z})$, $\boldsymbol{G}(\boldsymbol{z}')$ are generated. $\boldsymbol{z}$ is sampled from $\mathcal{Z}$ and $\boldsymbol{z}' = \boldsymbol{z} + \boldsymbol{A}(d, \varepsilon)$, where $d \in \{1, ..., D\}$ and $\varepsilon \in \mathbb{R}$, and $\boldsymbol{A}(d, \varepsilon)$ denotes the shift along the $d$-th direction with $\varepsilon$ scalar. 3) The $\Delta$-*Contrastor*, composed of two shared-weight Disentangling Encoders $\boldsymbol{E}$, encodes the image pair to a sample $\boldsymbol{v} \in \mathcal{V}$ as

$$\boldsymbol{v}(\boldsymbol{z}, d, \varepsilon) = |\boldsymbol{E}(\boldsymbol{G}(\boldsymbol{z} + \boldsymbol{A}(d, \varepsilon))) - \boldsymbol{E}(\boldsymbol{G}(\boldsymbol{z}))|, \tag{1}$$

where $\mathcal{V} \in \mathbb{R}_+^J$ denotes the *Variation Space*. Then we apply Contrastive Learning in $\mathcal{V}$ to optimize the Disentangling Encoder $\boldsymbol{E}$ to extract disentangled representations and simultaneously enable *Navigator* $\boldsymbol{A}$ to find the disentangled directions in the latent space $\mathcal{Z}$.

## 3.2 DESIGN OF DisCo

We present the design details of DisCo, which include: ($i$) the collection of query set $\mathcal{Q} = \{\boldsymbol{q}_i\}_{i=1}^B$, positive key set $\mathcal{K}^+ = \{\boldsymbol{k}_i^+\}_{i=1}^N$ and negative key set $\mathcal{K}^- = \{\boldsymbol{k}_i^-\}_{i=1}^M$, which are three subsets of the *Variation Space* $\mathcal{V}$, ($ii$) the formulation of the Contrastive Loss.

According to our goal of contrasting the variations, the samples from $\mathcal{Q}$ and $\mathcal{K}^+$ share the same traversal direction and should be pulled together, while the samples from $\mathcal{Q}$ and $\mathcal{K}^-$ have different directions and should be pushed away. Recall that each sample $\boldsymbol{v}$ in $\mathcal{V}$ is determined as $\boldsymbol{v}(\boldsymbol{z}, d, \varepsilon)$. To achieve the contrastive learning process, we construct the query sample $\boldsymbol{q}_i = \boldsymbol{v}(\boldsymbol{z}_i, d_i, \varepsilon_i)$, the key sample $\boldsymbol{k}_i^+ = \boldsymbol{v}(\boldsymbol{z}_i^+, d_i^+, \varepsilon_i^+)$ and the negative sample $\boldsymbol{k}_i^- = \boldsymbol{v}(\boldsymbol{z}_i^-, d_i^-, \varepsilon_i^-)$. Specifically, we randomly sample a direction index $\hat{d}$ from a discrete uniform distribution $\mathcal{U}\{1, D\}$ for $\{d_i\}_{i=1}^B$ and $\{d_i^+\}_{i=1}^N$ to guarantee they are the same. We randomly sample $\{d_i^-\}_{i=1}^M$ from the set of the rest of the directions $\mathcal{U}\{1, D\} \setminus \{\hat{d}\}$ individually and independently to cover the rest of directions in Navigator $\boldsymbol{A}$. Note that the discovered direction should be independent with the starting point and the scale of variation, which is in line with the disentangled factors. Therefore, $\{\boldsymbol{z}_i\}_{i=1}^B$, $\{\boldsymbol{z}_i^+\}_{i=1}^N$, $\{\boldsymbol{z}_i^-\}_{i=1}^M$ are all sampled from latent space $\mathcal{Z}$, and $\{\varepsilon_i\}_{i=1}^B$, $\{\varepsilon_i^+\}_{i=1}^N$, $\{\varepsilon_i^-\}_{i=1}^M$ are all sampled from a shared continuous uniform distribution $\mathcal{U}[-\epsilon, \epsilon]$ individually and independently. We normalize each sample in $\mathcal{Q}$, $\mathcal{K}^+$, and $\mathcal{K}^-$ to a unit vector to eliminate the impact caused by different shift scalars.

For the design of Contrastive Loss, a well-known form of Contrastive Loss is InfoNCE (van den Oord et al., 2018):

$$\mathcal{L}_{NCE} = -\frac{1}{|B|} \sum_{i=1}^{B} \sum_{j=1}^{N} \log \frac{\exp(\boldsymbol{q}_i \cdot \boldsymbol{k}_j^+ / \tau)}{\sum_{s=1}^{N+M} \exp(\boldsymbol{q}_i \cdot \boldsymbol{k}_s / \tau)}, \tag{2}$$

where $\tau$ is a temperature hyper-parameter and $\{\boldsymbol{k}_i\}_{i=1}^{N+M} = \{\boldsymbol{k}_i^+\}_{i=1}^{N} \bigcup \{\boldsymbol{k}_i^-\}_{i=1}^{M}$. The InfoNCE is originate from BCELoss (Gutmann & Hyvärinen, 2010). BCELoss has been used to achieve contrastive learning (Wu et al., 2018; Le-Khac et al., 2020; Mnih & Kavukcuoglu, 2013; Mnih & Teh, 2012). We choose to follow them to use BCELoss $\mathcal{L}_{logits}$ for reducing computational cost:

$$\mathcal{L}_{logits} = -\frac{1}{|B|} \sum_{i=1}^{B} \left( l_i^- + l_i^+ \right), \tag{3}$$

$$l_i^+ = \sum_{j=1}^{N} \log \sigma(\boldsymbol{q}_i \cdot \boldsymbol{k}_j^+ / \tau), \quad l_i^- = \sum_{m=1}^{M} \log(1 - \sigma(\boldsymbol{q}_i \cdot \boldsymbol{k}_m^- / \tau)), \tag{4}$$

where $\sigma$ denotes the sigmoid function, $l_i^+$ denotes the part for positive samples, and $l_i^-$ denotes the part for the negative ones. Note that we use a shared positive set for $B$ different queries to reduce the computational cost.

## 3.3 KEY TECHNIQUES FOR DISCO

**Entropy-based domination loss.** By optimizing the Contrastive Loss, *Navigator* $\boldsymbol{A}$ is optimized to find the disentangled directions in the latent space, and Disentangling Encoder $\boldsymbol{E}$ is optimized to extract disentangled representations from images. To further make the encoded representations more disentangled, i.e., when traversing along one disentangled direction, only one dimension of the encoded representation should respond, we thus propose an entropy-based domination loss to encourage the corresponding samples in the *Variation Space* to be one-hot. To implement the entropy-based domination loss, we first get the mean $\boldsymbol{c}$ of $\mathcal{Q}$ and $\mathcal{K}^+$ as

$$\boldsymbol{c} = \frac{1}{|B + N|} \left( \sum_{i=1}^{B} \boldsymbol{q}_i + \sum_{i=1}^{N} \boldsymbol{k}_i^+ \right). \tag{5}$$

We then compute the probability as $p_i = \exp \boldsymbol{c}(i) / \sum_{j=1}^{J} \exp \boldsymbol{c}(j)$, where $\boldsymbol{c}(i)$ is the $i$-th element of $\boldsymbol{c}$ and $J$ is the number of dimensions of $\boldsymbol{c}$. The entropy-based domination loss $\mathcal{L}_{ed}$ is calculated as

$$\mathcal{L}_{ed} = -\frac{1}{J} \sum_{j=1}^{J} p_j \log(p_j). \tag{6}$$

**Hard negatives flipping.** Since the latent space of the generative models is a high-dimension complex manifold, many different directions carry the same semantic meaning. These directions with the same semantic meaning result in hard negatives during the optimization of Contrastive Loss. The hard negatives here are different from the hard negatives in the works of self-supervised representation learning (He et al., 2020; Coskun et al., 2018), where they have reliable annotations of the samples. Here, our hard negatives are more likely to be "false" negatives, and we choose to flip these hard negatives into positives. Specifically, we use a threshold $T$ to identify the hard negative samples, and use their similarity to the queries as the pseudo-labels for them:

$$\hat{l}_i^- = \sum_{\alpha_{ij} < T} \log(1 - \sigma(\alpha_{ij})) + \sum_{\alpha_{ij} \geq T} \alpha_{ij} \log(\sigma(\alpha_{ij})), \tag{7}$$

where $\hat{l}_i^-$ denotes the modified $l_i^-$, and $\alpha_{ij} = \boldsymbol{q}_i \cdot \boldsymbol{k}_j^- / \tau$. Therefore, the modified final BCELoss is:

$$\mathcal{L}_{logits-f} = \frac{1}{|B|} \sum_{i=1}^{B} \left( l_i^+ + \hat{l}_i^- \right). \tag{8}$$

| Method | Cars3D | | Shapes3D | | MPI3D | |
|---|---|---|---|---|---|---|
| | MIG | DCI | MIG | DCI | MIG | DCI |
| *Typical disentanglement baselines:* | | | | | | |
| FactorVAE | $0.142 \pm 0.023$ | $0.161 \pm 0.019$ | $0.434 \pm 0.143$ | $0.611 \pm 0.101$ | $0.099 \pm 0.029$ | $0.240 \pm 0.051$ |
| $\beta$-TCVAE | $0.080 \pm 0.023$ | $0.140 \pm 0.019$ | $0.406 \pm 0.175$ | $0.613 \pm 0.114$ | $0.114 \pm 0.042$ | $0.237 \pm 0.056$ |
| InfoGAN-CR | $0.011 \pm 0.009$ | $0.020 \pm 0.011$ | $0.297 \pm 0.124$ | $0.478 \pm 0.055$ | $0.163 \pm 0.076$ | $0.241 \pm 0.075$ |
| *Methods on pretrained GAN:* | | | | | | |
| LD | $0.086 \pm 0.029$ | $0.216 \pm 0.072$ | $0.168 \pm 0.056$ | $0.380 \pm 0.062$ | $0.097 \pm 0.057$ | $0.196 \pm 0.038$ |
| CF | $0.083 \pm 0.024$ | $0.243 \pm 0.048$ | $0.307 \pm 0.124$ | $0.525 \pm 0.078$ | $0.183 \pm 0.081$ | $\mathbf{0.318 \pm 0.014}$ |
| GS | $0.136 \pm 0.006$ | $0.209 \pm 0.031$ | $0.121 \pm 0.048$ | $0.284 \pm 0.034$ | $0.163 \pm 0.065$ | $0.229 \pm 0.042$ |
| DS | $0.118 \pm 0.044$ | $0.222 \pm 0.044$ | $0.356 \pm 0.090$ | $0.513 \pm 0.075$ | $0.093 \pm 0.035$ | $0.248 \pm 0.038$ |
| DisCo (ours) | $\mathbf{0.179 \pm 0.037}$ | $\mathbf{0.271 \pm 0.037}$ | $\mathbf{0.512 \pm 0.068}$ | $\mathbf{0.708 \pm 0.048}$ | $\mathbf{0.222 \pm 0.027}$ | $0.292 \pm 0.024$ |
| *Methods on pretrained VAE:* | | | | | | |
| LD | $0.030 \pm 0.025$ | $0.068 \pm 0.030$ | $0.040 \pm 0.035$ | $0.068 \pm 0.075$ | $0.024 \pm 0.026$ | $0.035 \pm 0.014$ |
| DisCo (ours) | $\mathbf{0.103 \pm 0.028}$ | $\mathbf{0.211 \pm 0.041}$ | $\mathbf{0.331 \pm 0.161}$ | $\mathbf{0.844 \pm 0.033}$ | $\mathbf{0.068 \pm 0.030}$ | $\mathbf{0.288 \pm 0.021}$ |
| *Methods on pretrained Flow:* | | | | | | |
| LD | $0.015 \pm 0.000$ | $0.029 \pm 0.000$ | $0.067 \pm 0.000$ | $0.211 \pm 0.000$ | $0.025 \pm 0.000$ | $0.035 \pm 0.000$ |
| DisCo (ours) | $\mathbf{0.060 \pm 0.000}$ | $\mathbf{0.199 \pm 0.000}$ | $\mathbf{0.150 \pm 0.000}$ | $\mathbf{0.525 \pm 0.000}$ | $\mathbf{0.076 \pm 0.000}$ | $\mathbf{0.264 \pm 0.000}$ |

Table 1: Comparisons of the MIG and DCI disentanglement metrics (mean $\pm$ std). A higher mean indicates a better performance. DisCo can extract disentangled representations from all three generative models, and DisCo on GAN achieves the highest score in almost all the cases, compared to all the baselines. All the cells except for Flow are results over 25 runs.

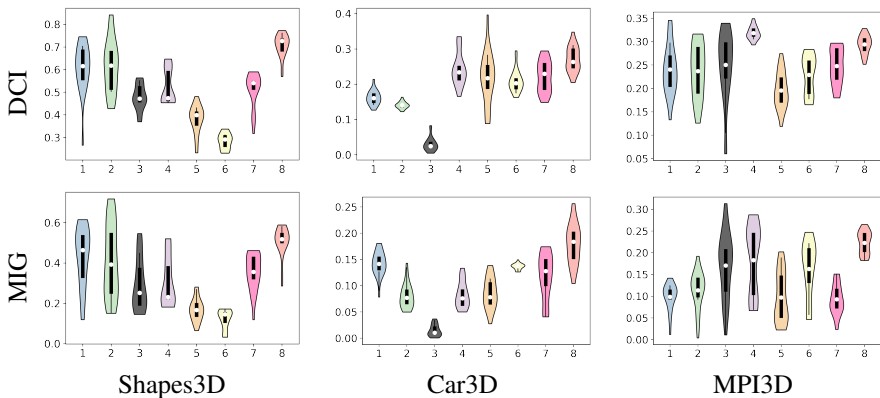

Figure 3: Violin plots on three datasets (1: $\beta$-TCVAE, 2: FactorVAE, 3: InfoGAN-CR, 4: CF, 5: LD, 6: GS, 7: DS, 8: DisCo (ours)). DisCo on pretrained GAN consistently achieves the best performance. Each method has 25 runs, and the variance is due to randomness.

**Full objective.** With the above two techniques, the full objective is:

$$\mathcal{L} = \mathcal{L}_{logits-f} + \lambda \mathcal{L}_{ed}, \tag{9}$$

where $\lambda$ is the weighting hyper-parameter for entropy-based domination loss $\mathcal{L}_{ed}$.

## 4 EXPERIMENT

In this section, we first follow the well-accepted protocol (Locatello et al., 2019; Khrulkov et al., 2021) to evaluate the learned disentangled representation, which also reflects the performance of discovered directions implicitly (Lin et al., 2020) (Section 4.1). Secondly, we follow Li et al. (2021a) to directly evaluate the discovered directions (Section 4.2). Finally, we conduct ablation study (Section 4.3).

### 4.1 EVALUATIONS ON DISENTANGLED REPRESENTATION

#### 4.1.1 EXPERIMENTAL SETUP

**Datasets.** We consider the following popular datasets in the disentanglement areas: **Shapes3D** (Kim & Mnih, 2018) with 6 ground truth factors, **MPI3D** (Gondal et al., 2019) with 7 ground truth factors,

LD    CF    DisCo

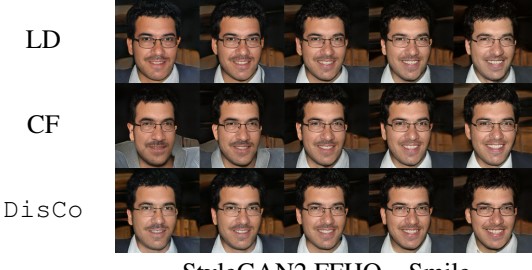


StyleGAN2 FFHQ – Smile          StyleGAN2 FFHQ – Bald

Figure 4: Comparison of discovered directions. `DisCo` can better manipulate desirable attributes while keeping others intact. Please refer to Appendix C for more qualitative results.

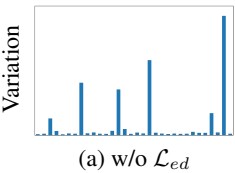
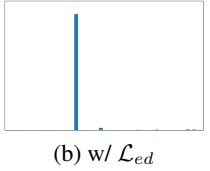

(a) w/o $\mathcal{L}_{ed}$        (b) w/ $\mathcal{L}_{ed}$

| Concat | Variation | Contrast | Classification | MIG | DCI |
|--------|-----------|----------|----------------|-------|-------|
| ✓ | | ✓ | | 0.023 | 0.225 |
| | ✓ | ✓ | | **0.562** | **0.736** |
| ✓ | | | ✓ | 0.012 | 0.138 |
| | ✓ | | ✓ | 0.002 | 0.452 |

Figure 5: Visualization of the variation of the encoded disentangled representations caused by the change of a single ground truth factor.

Table 2: Ablation on Contrast v.s. Classification and Concatenation (Concat) v.s. Variation.

and **Cars3D** (Reed et al., 2015) with 3 ground truth factors. In the experiments of the above datasets, images are resized to the 64x64 resolution.

**Pretrained generative models.** For GAN, we use the StyleGAN2 model (Karras et al., 2020). For VAE, we use a common structure with convolutions (Locatello et al., 2019). For Flow, we use Glow (Kingma & Dhariwal, 2018).

**Baseline.** For the typical disentanglement baselines, we choose **FactorVAE** (Kim & Mnih, 2018), $\beta$-**TCVAE** (Chen et al., 2018) and **InfoGAN-CR** (Lin et al., 2020). For discovering-based methods, we consider serveral recent methods: **GANspace (GS)** (Härkönen et al., 2020), **LatentDiscovery (LD)** (Voynov & Babenko, 2020), **ClosedForm (CF)** (Shen & Zhou, 2021) and **DeepSpectral (DS)** (Khrulkov et al., 2021). For these methods, we follow Khrulkov et al. (2021) to train an additional encoder to extract disentangled representation. We are the first to extract disentangled representations from pretrained VAE and Flow, so we extend **LD** to VAE and Flow as a baseline.

**Disentanglement metrics.** We mainly consider two representative ones: *the Mutual Information Gap (MIG)* (Chen et al., 2018) and the *Disentanglement metric (DCI)* (Eastwood & Williams, 2018). *MIG* requires each factor to be only perturbed by changes of a single dimension of representation. *DCI* requires each dimension only to encode the information of a single dominant factor. We evaluate the disentanglement in terms of both representation and factors. We also provide results for $\beta$-*VAE score* (Higgins et al., 2017) and *FactorVAE score* (Kim & Mnih, 2018) in Appendix B.3.

**Randomness.** We consider the randomness caused by random seeds and the strength of the regularization term (Locatello et al., 2019). For random seeds, we follow the same setting as the baselines. Since `DisCo` does not have a regularization term, we consider the randomness of the pretrained generative models. For all methods, we ensure there are 25 runs, except that Glow only has one run, limited by GPU resources. More details are presented in Appendix A.

### 4.1.2 EXPERIMENTAL RESULTS

The quantitative results are summarized in Table 1 and Figure 3. More details about the experimental settings and results are presented in Appendix A & C.

**`DisCo` vs. typical baselines.** Our `DisCo` achieves the SOTA performance consistently in terms of MIG and DCI scores. The variance due to randomness of `DisCo` tends to be smaller than those typical baselines. We demonstrate that the method, which extracts disentangled representation from pretrained non-disentangled models, can outperform typical disentanglement baselines.

**`DisCo` vs. discovering-based methods.** Among the baselines based on discovering pretrained GAN, **CF** achieves the best performance. `DisCo` outperforms **CF** in almost all the cases by a large margin. Besides, these baselines need an extra stage (Khrulkov et al., 2021) to get disentangled representation, while our Disentangling Encoder can directly extract disentangled representation.

## 4.2 EVALUATIONS ON DISCOVERED DIRECTIONS

To evaluate the discovered directions, we compare `DisCo` on StyleGAN2 with **GS**, **LD**, **CF** and **DS** on the real-world dataset FFHQ (Karras et al., 2019)[1]. and adopt the comprehensive Manipulation Disentanglement Score (MDS) (Li et al., 2021a) as a metric. To calculate MDS, we use $40$ CelebaHQ-Attributes predictors released by StyleGAN. Among them, we select **Young**, **Smile**, **Bald** and **Blonde Hair**, as they are attributes with an available predictor and commonly found by all methods at the same time. The results are summarized in Table 3. `DisCo` has shown better overall performance compared to other baselines, which verifies our assumption that learning disentangled representation benefits latent space discovering. We also provide qualitative comparisons in Figure 4.

| Method | MDS on CelebAHQ-Attributes | | | | |
| | Young | Smile | Bald | Blonde Hair | Overall |
|---|---|---|---|---|---|
| DS | 0.518 | 0.570 | 0.524 | 0.511 | 0.531 |
| CF | 0.518 | 0.553 | 0.504 | 0.560 | 0.534 |
| GS | 0.502 | 0.534 | 0.494 | 0.538 | 0.517 |
| LD | **0.627** | 0.531 | 0.524 | 0.514 | 0.549 |
| DisCo | 0.516 | **0.688** | **0.568** | **0.592** | **0.591** |

Table 3: MDS comparison on facial attribute editing. Our `DisCo` shows the best overall score for the latent discovering task on FFHQ dataset.

Finally, we provide an intuitive analysis in Appendix D for why `DisCo` can find those disentangled directions.

## 4.3 ABLATION STUDY

In this section, we perform ablation study of `DisCo` only on GAN, limited by the space. For the experiments, we use the Shapes3D dataset, and the random seed is fixed.

**Choice of latent space.** For style–based GANs (Karras et al., 2019; 2020), there is a style space $\mathcal{W}$, which is the output of style network (MLP) whose input is a random latent space $\mathcal{Z}$. As demonstrated in Karras et al. (2019), $\mathcal{W}$ is more interpretable than $\mathcal{Z}$. We conduct experiments on $\mathcal{W}$ and $\mathcal{Z}$ respectively to see how the latent space influences the performance. As shown in Table 4, `DisCo` on $\mathcal{W}$ is better, indicating that the better the latent space is organized, the better disentanglement `DisCo` can achieve.

**Choices of $A$.** Following the setting of Voynov & Babenko (2020), we mainly consider three options of $A$: a linear operator with all matrix columns having a unit length, a linear operator with orthonormal matrix columns, or a non-linear operator of 3 fully-connected layers.

The results are shown in Table 4. For latent spaces $\mathcal{W}$ and $\mathcal{Z}$, $A$ with unit-norm columns achieves nearly the best performance in terms of MIG and DCI scores. Compared to $A$ with orthonormal matrix columns, using $A$ with unit-norm columns is more expressive with less constraints. Another possible reason is that $A$ is global without conditioned on the latent code $z$. A non-linear operator is more suitable for a local navigator $A$. For such a much more complex local and non-linear setting, more inductive bias or supervision should be introduced.

| Method | MIG | DCI |
|---|---|---|
| $\mathcal{Z}$ + Unit length matrix | **0.242** | **0.673** |
| $\mathcal{Z}$ + Orthonormal matrix | 0.183 | 0.578 |
| $\mathcal{Z}$ + 3 fully-connected layers | 0.169 | 0.504 |
| $\mathcal{W}$ + Unit length matrix | 0.547 | **0.730** |
| $\mathcal{W}$ + Orthonormal matrix | **0.551** | 0.709 |
| $\mathcal{W}$ + 3 fully-connected layers | 0.340 | 0.665 |
| $\mathcal{L}_{logits}$ | 0.134 | 0.632 |
| $\mathcal{L}_{logits} + \mathcal{L}_{ed}$ | 0.296 | 0.627 |
| $\mathcal{L}_{logits-f}$ | 0.134 | 0.681 |
| $\mathcal{L}_{logits-f} + \mathcal{L}_{ed}$ | **0.547** | **0.730** |

Table 4: Ablation study of `DisCo` on the latent spaces, types of $A$, and our proposed techniques.

**Entropy-based domination loss.** Here, we verify the effectiveness of entropy-based domination loss $\mathcal{L}_{ed}$ for disentanglement. For a desirable disentangled representation, one semantic meaning corresponds to one dimension. As shown in Table 4, $\mathcal{L}_{ed}$ can improve the performance by a large

---

[1]The above disentanglement metrics (DCI and MIG) are not available for FFHQ dataset.

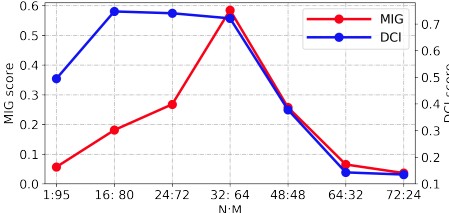 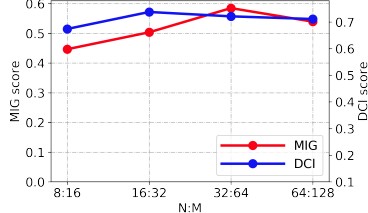

(a) Impact of $N : M$ with a fixed sum     (b) Impact of $N + M$ with a fixed ratio $1 : 2$

Figure 6: Study on numbers of positive (N) and negative samples (M). The balance between positive and negative samples is crucial for `DisCo`.

margin. We also visualize the *Variation Space* to further demonstrate the effectiveness of our proposed loss in Figure 5. Adding the domination loss makes the samples in the *Variation Space* to be one-hot, which is desirable for disentanglement.

**Hard negatives flipping.** We run our `DisCo` with or without the hard negatives flipping strategy to study its influence. As shown in Table 4, flipping hard negatives can improve the disentanglement ability of `DisCo`. The reason is that the hard negatives have the same semantics as the positive samples. In this case, treating them as the hard negatives does not make sense. Flipping them with pseudo-labels can make the optimization of Contrastive Learning easier.

**Hyperparmeter N & M.** We run `DisCo` with different ratios of $N : M$ with a fixed sum of 96, and different sum of $N + M$ with a fixed ratio $1 : 2$ to study their impacts. As shown in Figure 6 (a), the best ratio is $N : M = 32 : 64 = 1 : 2$, as the red line (MIG) and blue line (DCI) in the figure show that larger or smaller ratios will hurt `DisCo`, which indicates `DisCo` requires a balance between $N$ and $M$. As shown in Figure 6 (b), the sum of $N + M$ has slight impact on `DisCo`. For other hyperparameters, we set them empirically, and more details are presented in Appendix A.

**Contrast vs. Classification.** To verify the effectiveness of Contrast, we substitute it with classification by adopting an additional linear layer to recover the corresponding direction index and the shift along this direction. As Table 2 shows, Contrastive Learning outperforms Classification significantly.

**Concatenation vs. Variation.** We further demonstrate that the *Variation Space* is crucial for `DisCo`. By replacing the difference operator with concatenation, the performance drops significantly (Table 2), indicating that the encoded representation is not well disentangled. On the other hand, the disentangled representations of images are achieved by Contrastive Learning in the *Variation Space*.

## 4.4 ANALYSIS OF DIFFERENT GENERATIVE MODELS

As shown in Table 1, `DisCo` can be well generalized to different generative models (GAN, VAE, and Flow). `DisCo` on GAN and VAE can achieve relative good performance, while `DisCo` on Flow is not as good. The possible reason is similar to the choice of latent space of GAN. We assume the disentangled directions are global linear and thus use a linear navigator. In contrast to GAN and VAE, we suspect that Flow may not conform to this assumption well. Furthermore, Flow has the problems of high GPU cost and unstable training, which limit us to do further exploration.

## 5 CONCLUSION

In this paper, we present an unsupervised and model-agnostic method `DisCo`, which is a Contrastive Learning framework to learn disentangled representation by exploiting pretrained generative models. We propose an entropy-based domination loss and a hard negatives flipping strategy to achieve better disentanglement. `DisCo` outperforms typical unsupervised disentanglement methods while maintaining high image quality. We pinpoint a new direction that Contrastive Learning can be well applied to extract disentangled representation from pretrained generative models. There may be some specific complex generative models, for which the global linear assumption of disentangled directions in the latent space could be a limitation. For future work, extending `DisCo` to the existing VAE-based disentanglement framework is an exciting direction.

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

# A  IMPLEMENTATION DETAILS

## A.1  SETTING FOR `DisCo`

For the hyperparameters, we empirically set the temperature $\tau$ to 1, threshold $T$ to 0.95, batch size $B$ to 32, the number of positives $N$ to 32, the number of negatives $K$ to 64, the loss weight $\lambda$ for $\mathcal{L}_{ed}$ to 1, the number of directions $D$ to 64 and the dimension of the representation $J$ to 32. We use an Adam optimizer (Kingma & Ba, 2015) in the training process, as shown in Table 5. Besides $N$ and $M$, we empirically find that `DisCo` is not sensitive to threshold $T \geq 0.9$ and other hyperparameters.

For the randomness, there is no regularization term for `DisCo`, thus the disentanglement performance is mainly influenced by the pretrained generative models. We follow Khrulkov et al. (2021) to run 5 random seeds to pretrain the GAN and 5 random seeds for training `DisCo`. We have the same setting for `DisCo` on GAN and VAE on all three datasets. For Flow, we only use one random seed to pretrain the Glow and use one random seed for `DisCo`. Compare with the baselines, for DisCo on StyleGAN, our modification happens globally on all layers in the $W$ space without any manual selection.

| Parameter | Values |
|---|---|
| Optimizer | Adam |
| Adam: beta1 | 0.9 |
| Adam: beta2 | 0.999 |
| Adam: epsilon | 1.00e-08 |
| Adam: learning rate | 0.00001 |
| Iteration: | 100,000 |

Table 5: Optimizer for `DisCo`.

| | $T = 0.7$ | $T = 0.8$ | $T = 0.9$ | $T = 0.95$ | $T = 0.98$ |
|---|---|---|---|---|---|
| MIG | 0.157 | 0.244 | 0.508 | **0.547** | 0.408 |
| DCI | 0.396 | 0.576 | 0.710 | **0.730** | 0.703 |

Table 6: Ablation study on hyperparameter $T$. `DisCo` is not sensitive to $T$ when $T \geq 0.9$. For $T < 0.9$, we may flip true hard negative and thus lead the optimization of Contrastive Loss collapse.

## A.2 SETTING FOR BASELINES

In this section, we introduce the implementation setting for the baselines (including randomness).

**VAE-based methods.** We choose FactorVAE and $\beta$-TCVAE as the SOTA VAE-based methods, we follow Locatello et al. (2019) to use the same architecture of encoder and decoder. For the **hyper-parameter**s, we use the the best settings by grid search. We set the latent dimension of representation to 10. For FactorVAE, we set the hyperparameter $\gamma$ to 10. For $\beta$-TCVAE, we set the hyperparameter $\beta$ to 6. For the **random seed**s, considering our method has 25 run, we run 25 times with different random seeds for each model to make the comparison fair.

**InfoGAN-based methods.** We choose InfoGAN-CR as a baseline. We use the official implementation [2] with the best **hyperparameter** settings by grid search. For the **random seed**s, we run 25 times with different random seeds

**Discovering-based methods.** We follow Khrulkov et al. (2021) to use the same settings for the following four baselines: **LD** (GAN), **CF**, **GS**, and **DS**. Similar to our method (`DisCo`), discovering-based methods do not have a regularization term. Thus, for the randomness, we adopt the same strategy with `DisCo`. We take the top-10 directions for 5 different random seeds for GAN and 5 different random seeds for the additional encoder to learn disentangled representations.

**LD (VAE) & LD (Flow).** We follow LD (GAN) to use the same settings and substitute the GAN with VAE / Glow. The only exception is the randomness for LD (Flow). We only run one random seed to pretrain the Glow and use one random seed for the encoder.

## A.3 MANIPULATION DISENTANGLEMENT SCORE

As claimed in Li et al. (2021a), it is difficult to evaluate the performance on discovering the latent space among different methods, which often use model-specific hyper-parameters to control the editing strength. Thus, Li et al. (2021a) propose a comprehensive metric called **Manipulation Disentanglement Score** (MDS), which takes both the accuracy and the disentanglement of manipulation into consideration. For more details, please refer to Li et al. (2021a).

## A.4 DOMAIN GAP PROBLEM

Please note that there exists a domain gap between the generated images of pretrained generative models and the real images. However, the good performance on disentanglement metrics shows that the domain gap has limited influence on `DisCo`.

---

[2] `https://github.com/fjxmlzn/InfoGAN-CR`

## A.5 ARCHITECTURE

Here, we provide the model architectures in our work. For the architecture of StyleGAN2, we follow Khrulkov et al. (2021). For the architecture of Glow, we use the open-source implementation [3].

| |
| --- |
| Conv $7 \times 7 \times 3 \times 64$, `stride` $= 1$ |
| ReLu |
| Conv $4 \times 4 \times 64 \times 128$, `stride` $= 2$ |
| ReLu |
| Conv $4 \times 4 \times 128 \times 256$, `stride` $= 2$ |
| ReLu |
| Conv $4 \times 4 \times 256 \times 256$, `stride` $= 2$ |
| ReLu |
| Conv $4 \times 4 \times 256 \times 256$, `stride` $= 2$ |
| ReLu |
| FC $4096 \times 256$ |
| ReLu |
| FC $256 \times 256$ |
| ReLu |
| FC $256 \times J$ |

Table 7: Encoder $E$ architecture used in `DisCo`. $J$ is 32 for Shapes3D, MPI3D and Car3D.

| |
| --- |
| FC $J \times 256$ |
| ReLu |
| FC $256 \times 256$ |
| ReLu |
| FC $256 \times 4096$ |
| ConvTranspose $4 \times 4 \times 256 \times 256$, `stride` $= 2$ |
| ReLu |
| ConvTranspose $4 \times 4 \times 256 \times 256$, `stride` $= 2$ |
| ReLu |
| ConvTranspose $4 \times 4 \times 256 \times 128$, `stride` $= 2$ |
| ReLu |
| ConvTranspose $4 \times 4 \times 128 \times 64$, `stride` $= 2$ |
| ReLu |
| ConvTranspose $7 \times 7 \times 64 \times 3$, `stride` $= 1$ |

Table 8: VAE's decoder architecture. Its encoder is the same as the encoder in `DisCo`.

---

[3] `https://github.com/rosinality/glow-pytorch`

# B  MORE EXPERIMENTS

## B.1  MORE QUALITATIVE COMPARISON

We provide some examples for qualitative comparison. We first demonstrate the trade-off problem of the VAE-based methods. As shown in Figure 7, `DisCo` leverages the pretrained generative model and does not have the trade-off between disentanglement and generation quality.

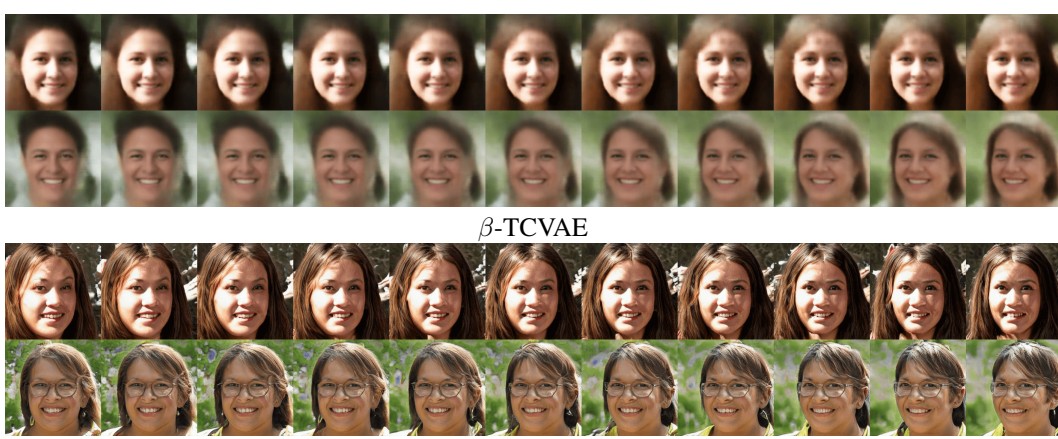

$\beta$-TCVAE

`DisCo`

Figure 7: Demonstration of the **trade-off** problem of the VAE-based method. $\beta$-TCVAE has bad generation quality, especially on the real-world dataset. `DisCo` lerverages pretrained generative model that can synthesize high-quality images.

Furthermore, as shown in Figure 8 and Figure 9, VAE-based methods suffer from poor image quality. When changing one attribute, the results of discovering-based methods tend to also change other attributes.

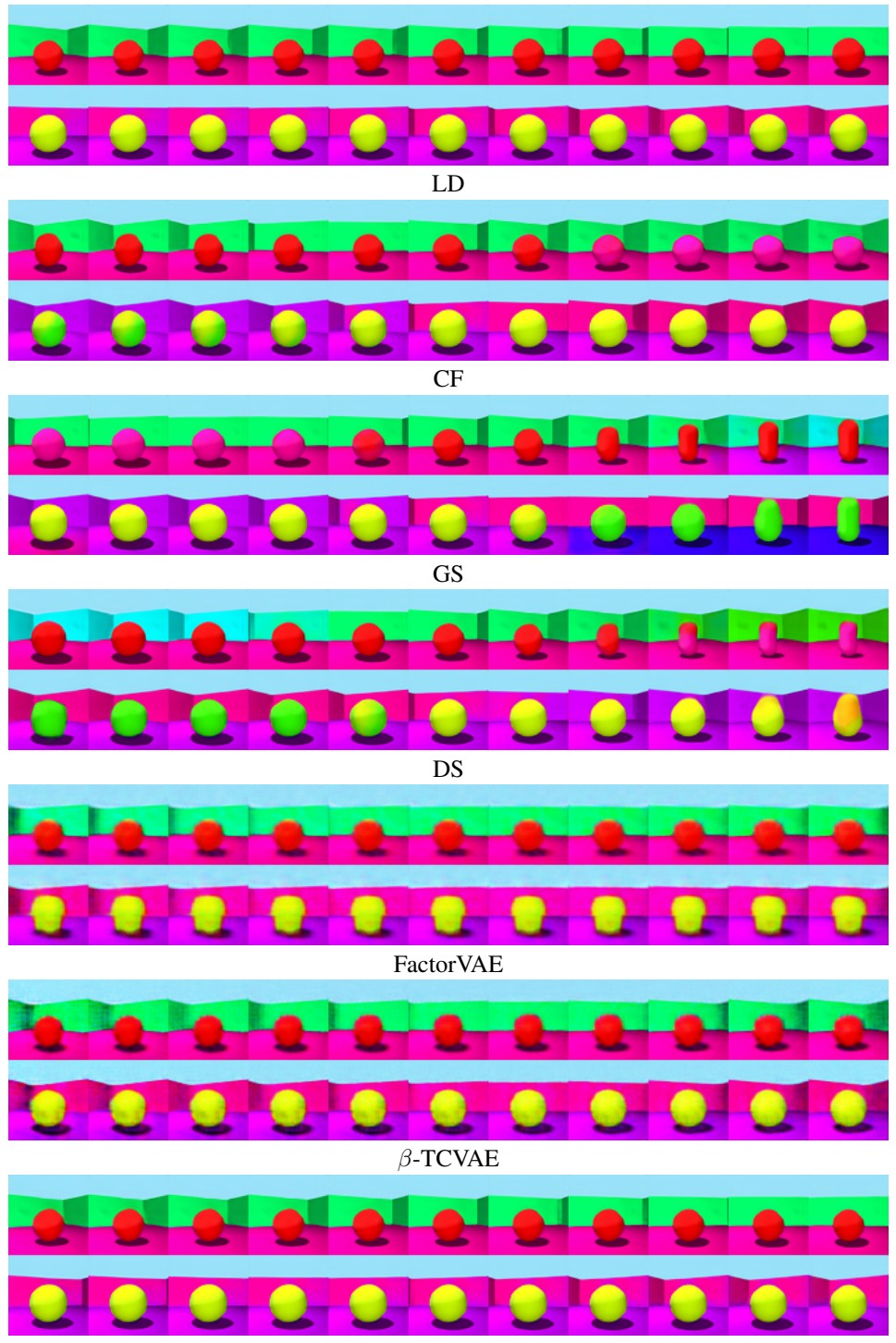

Figure 8: Comparison with baselines on Shapes3D dataset with *Pose* attribute.

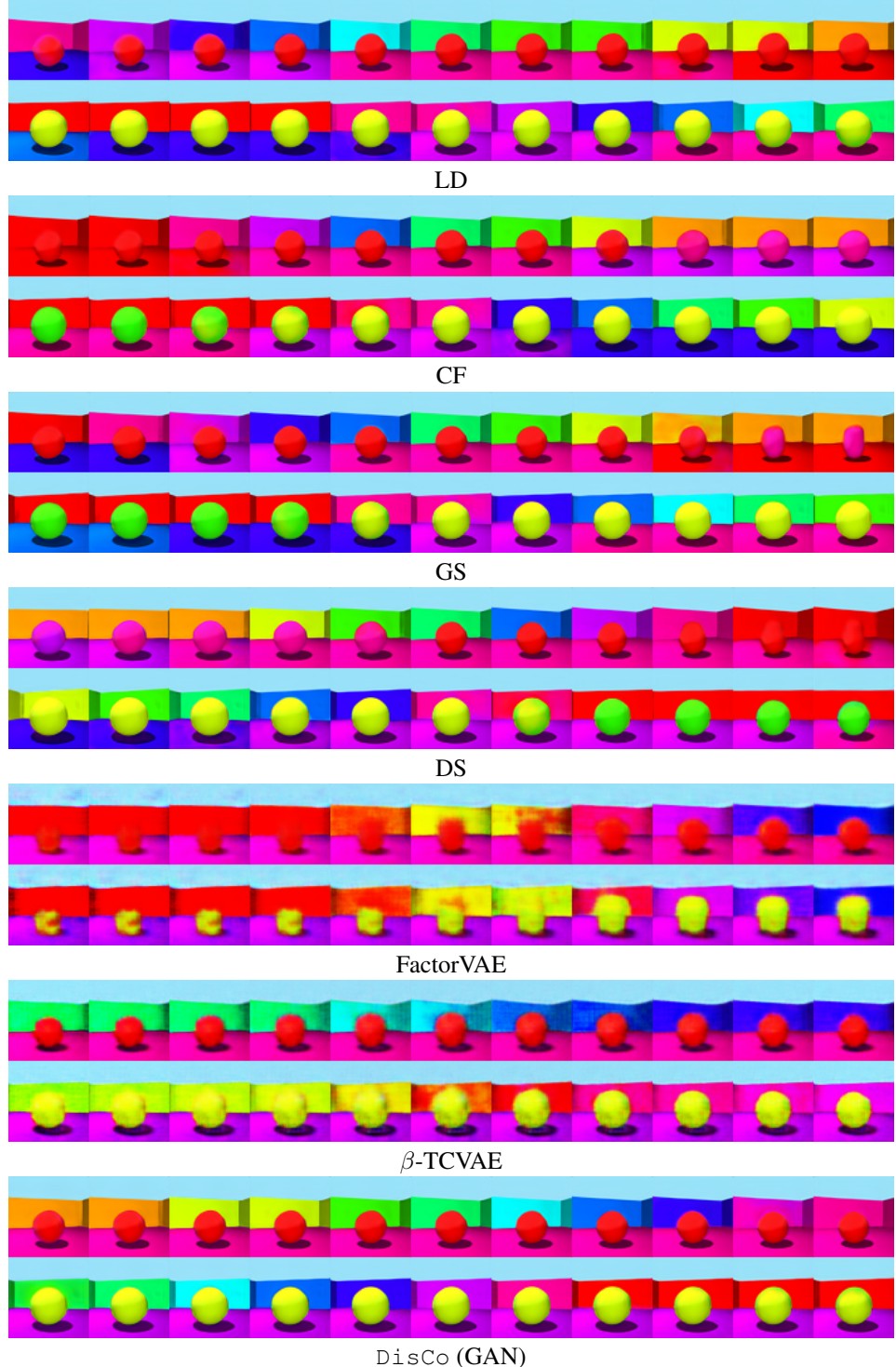

Figure 9: Comparison with baselines on Shapes3D dataset with *Wall Color* attribute. VAE-based methods suffer from poor image quality. Discovering-based methods tend to entangle *Wall Color* with other attributes.

We also provide qualitative comparisons between `DisCo` and InfoGAN-CR. Note that the latent space of InfoGAN-CR is not aligned with the pretrained StyleGAN2. InfoGAN-CR also suffers from the trade-off problem, and its disentanglement ability is worse than `DisCo`.

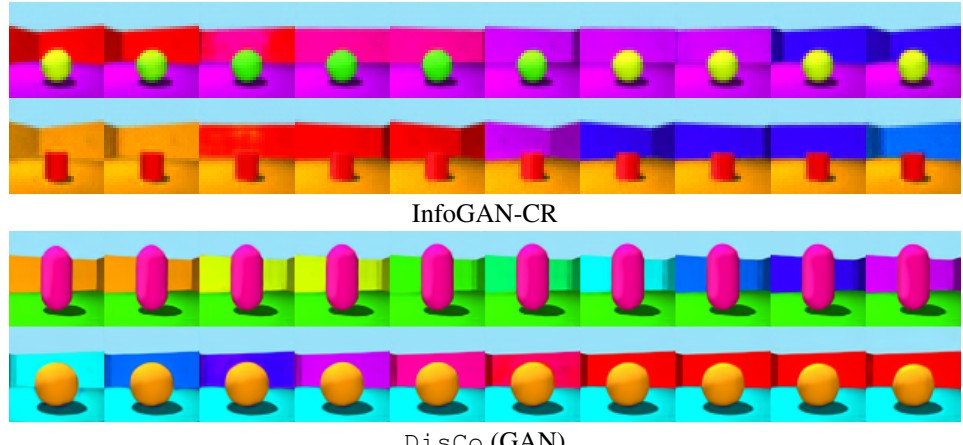

InfoGAN-CR

`DisCo` (GAN)

Figure 10: Comparison with baselines on Shapes3D dataset with *Wall Color* attribute. InfoGAN-CR entangles *Wall Color* with *Object Color* and *Pose*.

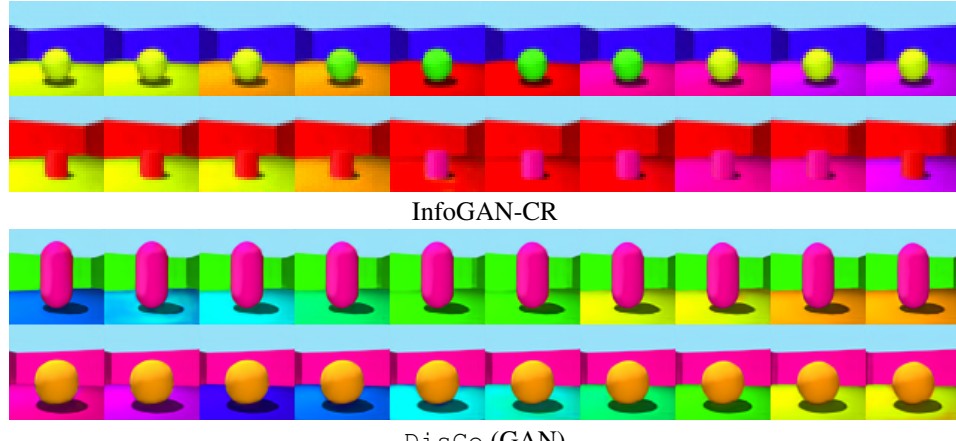

InfoGAN-CR

`DisCo` (GAN)

Figure 11: Comparison with baselines on Shapes3D dataset with *Floor Color* attribute. InfoGAN-CR entangles *Floor Color* with *Object Color*.

We explain the comparison in the main paper and show more manipulation comparisons here.

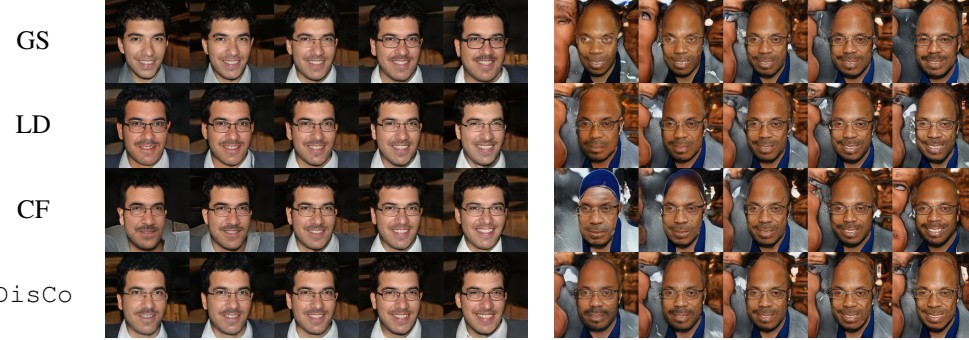

StyleGAN2 FFHQ – Smile

Figure 12: Manipulation comparison with discovering-based pipeline with *Smile* attribute. We explain the left column here. For **GS**, the manipulation also changes age. For **LD**, the manipulation also changes pose and skin tone. For **CF**, the manipulation also change identity.

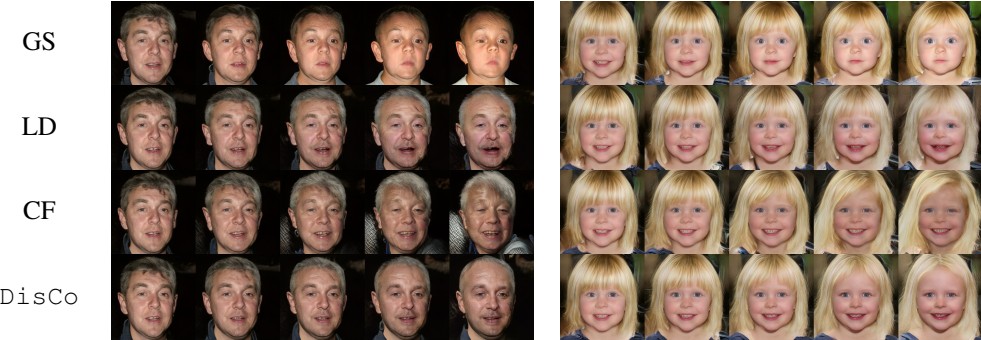

StyleGAN2 FFHQ – Bald

Figure 13: Manipulation comparison with discovering-based pipeline with *Bald* attribute. We explain the left column here. For **GS** and **LD**, the manipulations also change age. For **CF**, the manipulation also changes skin tone.

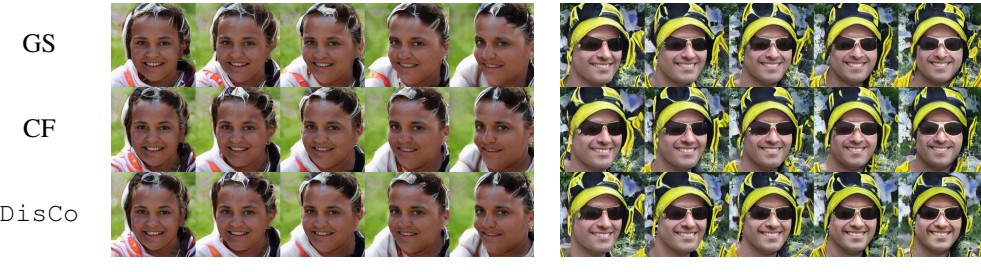

StyleGAN2 FFHQ – Pose

Figure 14: Manipulation comparison with discovering-based pipeline with *Pose* attribute. **LD** does not find the direction of pose attribute. **GS**, **CF** and `DisCo` can manipulate pose successfully.

## B.2 ANALYSIS OF THE LEARNED DISENTANGLED REPRESENTATIONS

We feed the images traversing the three most significant factors (wall color, floor color, and object color) of Shapes3D into the Disentangling Encoders and plot the corresponding dimensions of the encoded representations to visualize the learned disentangled space. The location of each point is the disentangled representation of the corresponding image. An ideal result is that all the points form a cube, and color variation is continuous. We consider three baselines that have relatively higher MIG and DCI: **CF**, **DS**, **LD**. As the figures below show, the points in the latent space of **CF** and **DS** are not well organized, and the latent space of all the three baselines are not well aligned with the axes, especially for **LD**. `DisCo` learns a well-aligned and well-organized latent space, which signifies a better disentanglement.

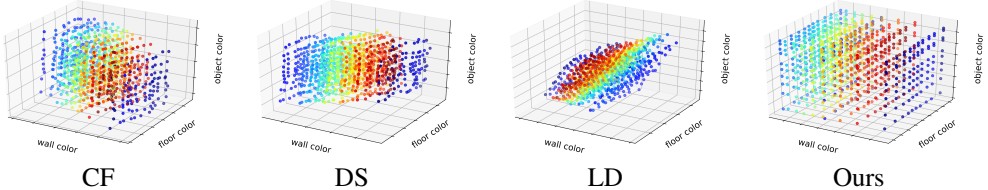

CF         DS         LD         Ours

## B.3 MORE QUANTITATIVE COMPARISON

We provide additional quantitative comparisons in terms of $\beta$-VAE score and FactorVAE score. `DisCo` on pretrained GAN is comparable to discovering-based baselines in terms of $\beta$-VAE score and FactorVAE score, suggesting that some disagreement between these two scores and MIG/ DCI. However, note that the qualitative evaluation in Figure 8, Figure 9 and Section B.2 shows that the disentanglement ability of `DisCo` is better than all the baselines on Shapes3D dataset.

| Method | Cars3D | | Shapes3D | | MPI3D | |
|---|---|---|---|---|---|---|
| | $\beta$-VAE score | FactorVAE score | $\beta$-VAE score | FactorVAE score | $\beta$-VAE score | FactorVAE score |
| *Typical disentanglement baselines:* | | | | | | |
| FactorVAE | $1.00 \pm 0.00$ | $0.906 \pm 0.052$ | $0.892 \pm 0.064$ | $0.840 \pm 0.066$ | $0.339 \pm 0.029$ | $0.152 \pm 0.025$ |
| $\beta$-TCVAE | $0.999 \pm 1.0e-4$ | $0.855 \pm 0.082$ | $0.978 \pm 0.036$ | $0.873 \pm 0.074$ | $0.348 \pm 0.012$ | $0.179 \pm 0.017$ |
| InfoGAN-CR | $0.450 \pm 0.022$ | $0.411 \pm 0.013$ | $0.837 \pm 0.039$ | $0.587 \pm 0.058$ | $0.672 \pm 0.101$ | $0.439 \pm 0.061$ |
| *Methods on pretrained GAN:* | | | | | | |
| LD | $0.999 \pm 2.54e-4$ | $0.852 \pm 0.039$ | $0.913 \pm 0.063$ | $0.805 \pm 0.064$ | $0.535 \pm 0.057$ | $0.391 \pm 0.039$ |
| CF | $1.00 \pm 0.00$ | $0.873 \pm 0.036$ | $0.999 \pm 0.001$ | $0.951 \pm 0.021$ | $0.669 \pm 0.033$ | $0.523 \pm 0.056$ |
| GS | $1.00 \pm 0.00$ | $0.932 \pm 0.018$ | $0.944 \pm 0.044$ | $0.788 \pm 0.091$ | $0.605 \pm 0.061$ | $0.465 \pm 0.036$ |
| DS | $1.00 \pm 0.00$ | $0.871 \pm 0.047$ | $0.991 \pm 0.022$ | $0.929 \pm 0.065$ | $0.651 \pm 0.043$ | $0.502 \pm 0.042$ |
| `DisCo` (ours) | $0.999 \pm 6.86e-5$ | $0.855 \pm 0.074$ | $0.987 \pm 0.028$ | $0.877 \pm 0.031$ | $0.530 \pm 0.015$ | $0.371 \pm 0.030$ |
| *Methods on pretrained VAE:* | | | | | | |
| LD | $0.951 \pm 0.074$ | $0.711 \pm 0.085$ | $0.602 \pm 0.196$ | $0.437 \pm 0.188$ | $0.266 \pm 0.068$ | $0.242 \pm 0.010$ |
| `DisCo` (ours) | $0.999 \pm 5.42e-5$ | $0.761 \pm 0.114$ | $0.999 \pm 8.9e-4$ | $0.956 \pm 0.041$ | $0.411 \pm 0.034$ | $0.391 \pm 0.075$ |
| *Methods on pretrained Flow:* | | | | | | |
| LD | $0.922 \pm 0.000$ | $0.633 \pm 0.000$ | $0.699 \pm 0.000$ | $0.597 \pm 0.000$ | $0.266 \pm 0.000$ | $0.242 \pm 0.000$ |
| `DisCo` (ours) | $1.00 \pm 0.000$ | $0.880 \pm 0.000$ | $0.860 \pm 0.000$ | $0.854 \pm 0.000$ | $0.538 \pm 0.000$ | $0.486 \pm 0.000$ |

Table 9: Comparisons of the $\beta$-VAE and FactorVAE scores on the Shapes3D dataset (mean $\pm$ variance). A higher mean indicates a better performance.

We also provide an additional experiment on Noisy-DSprites dataset. We compare `DisCo` with $\beta$-TCVAE (the best typical method) and CF (the best discovering-based method) in terms of MIG and DCI metrics.

| Method | $\beta$-TCVAE | CF | `DisCo` (GAN) |
|---|---|---|---|
| DCI | $0.088 \pm 0.049$ | $0.027 \pm 0.016$ | $\mathbf{0.120 \pm 0.059}$ |
| MIG | $0.046 \pm 0.031$ | $0.020 \pm 0.015$ | $\mathbf{0.104 \pm 0.030}$ |

Table 10: Comparisons on Noisy-DSprites.

## C   LATENT TRAVERSALS

In this section, we visualize the disentangled directions of the latent space discovered by `DisCo` on each dataset. For Cars3D, Shapes3D, Anime and MNIST, the iamge resolution is $64 \times 64$. For FFHQ, LSUN cat and LSUN church, the image resolution is $256 \times 256$. Besides StyleGAN2, we also provide results of Spectral Norm GAN (Miyato et al., 2018) [4] on MNIST (LeCun et al., 2010) and Anime Face (Jin et al., 2017) to demonstrate that `DisCo` can be well generalized to other types of GAN.

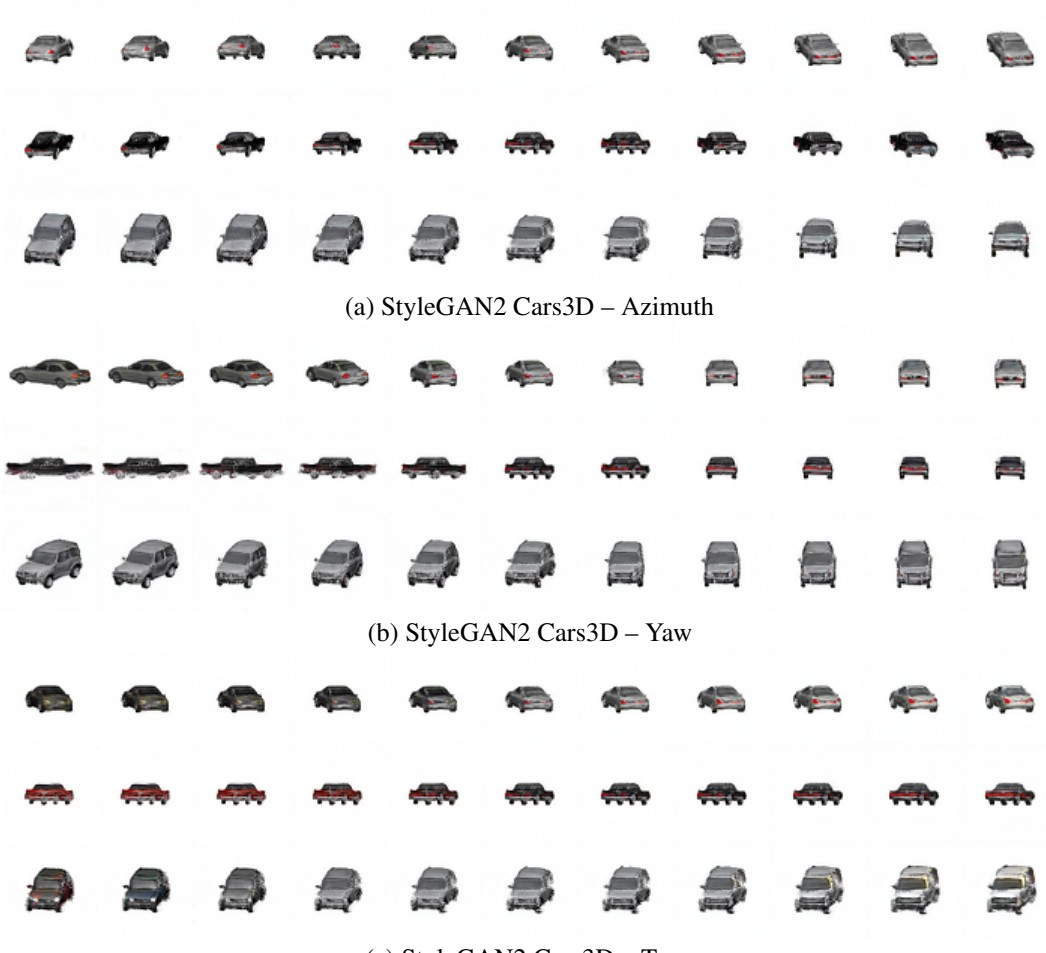

(a) StyleGAN2 Cars3D – Azimuth

(b) StyleGAN2 Cars3D – Yaw

(c) StyleGAN2 Cars3D – Type

Figure 15: Examples of disentangled directions for StyleGAN2 on Cars3D discovered by `DisCo`.

---

[4]`https://github.com/anvoynov/GANLatentDiscovery`

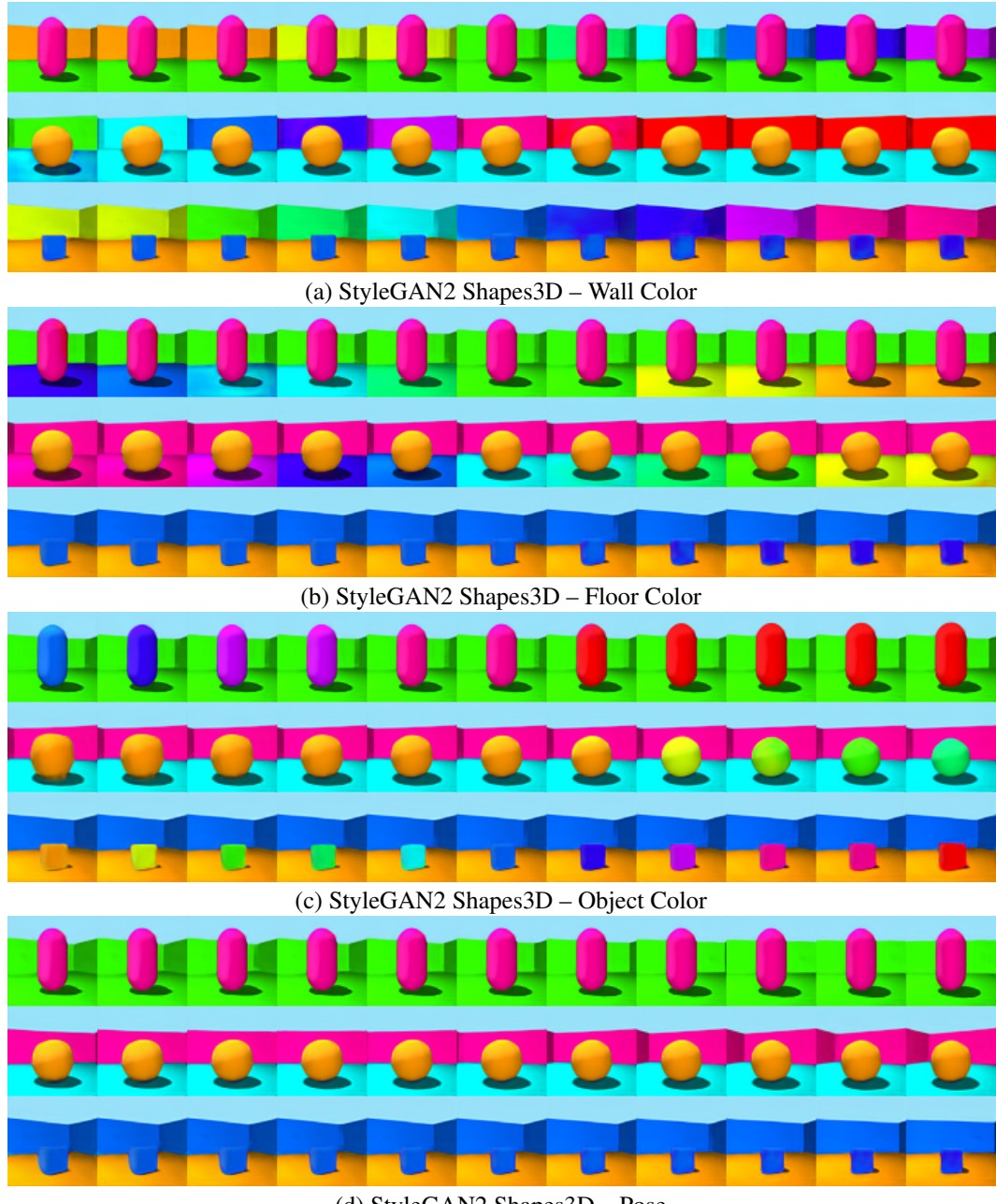

(a) StyleGAN2 Shapes3D – Wall Color

(b) StyleGAN2 Shapes3D – Floor Color

(c) StyleGAN2 Shapes3D – Object Color

(d) StyleGAN2 Shapes3D – Pose

Figure 16: Examples of disentangled directions for StyleGAN2 on Shapes3D discovered by `DisCo`. As shown in (b), the latent space has local semantic.

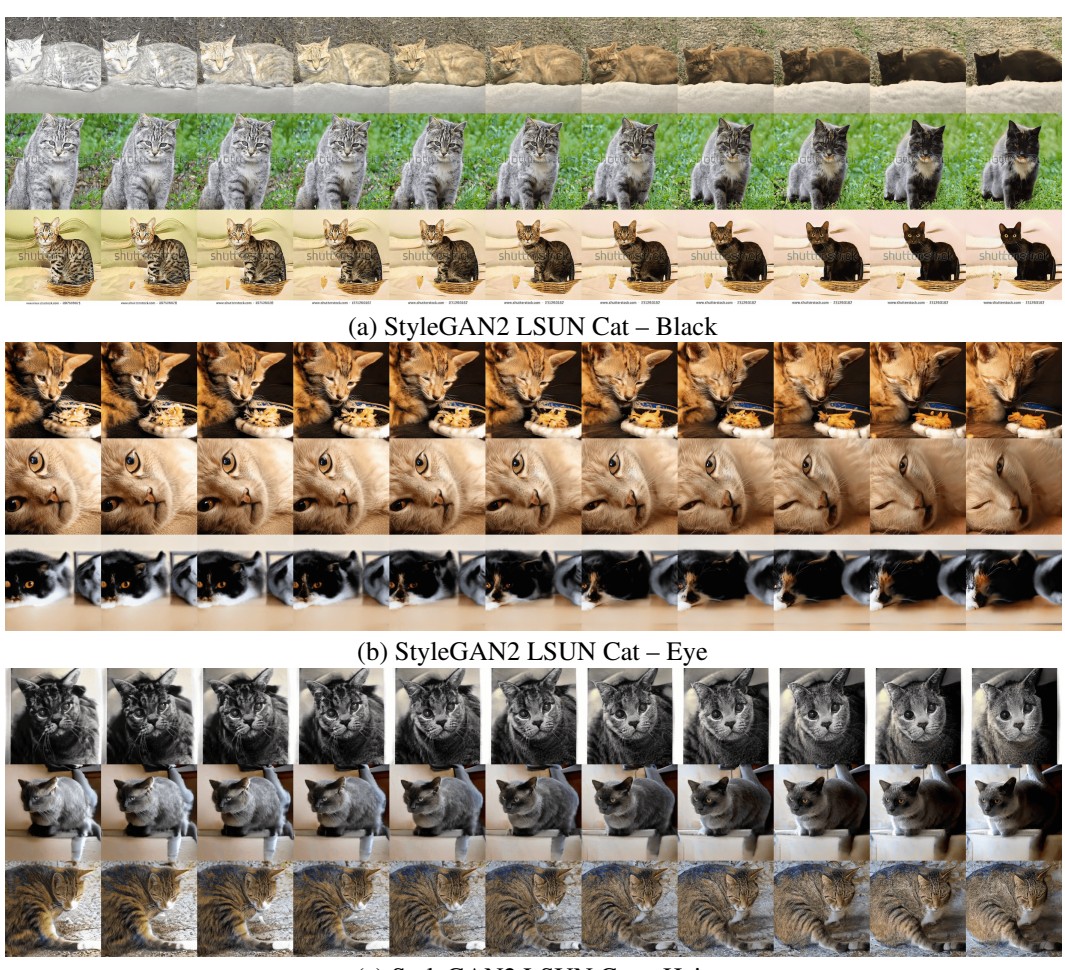

(a) StyleGAN2 LSUN Cat – Black

(b) StyleGAN2 LSUN Cat – Eye

(c) StyleGAN2 LSUN Cat – Hair

Figure 17: Examples of disentangled directions for StyleGAN2 on LSUN Cat discovered by `DisCo`.

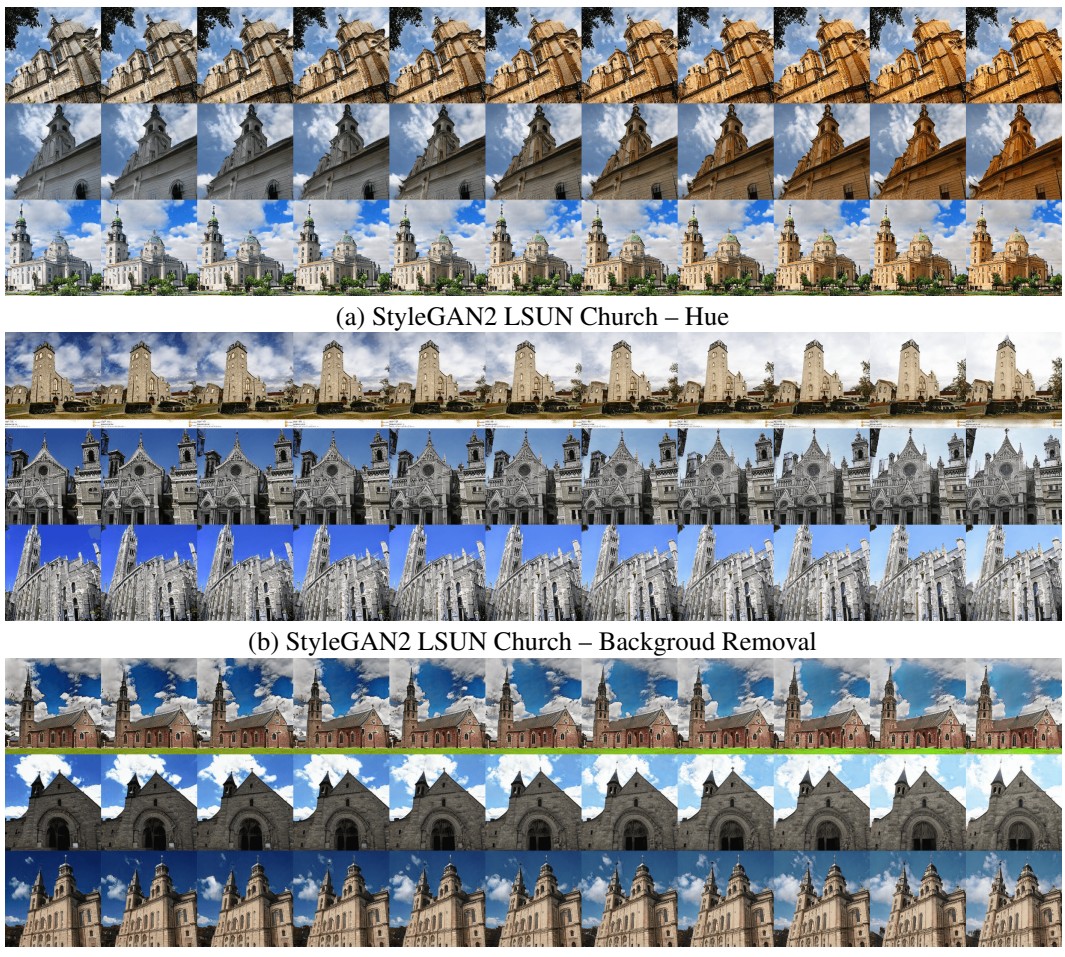

(a) StyleGAN2 LSUN Church – Hue

(b) StyleGAN2 LSUN Church – Backgroud Removal

(c) StyleGAN2 LSUN Church – Sky

Figure 18: Examples of disentangled directions for StyleGAN2 on LSUN Church discovered by `DisCo`.

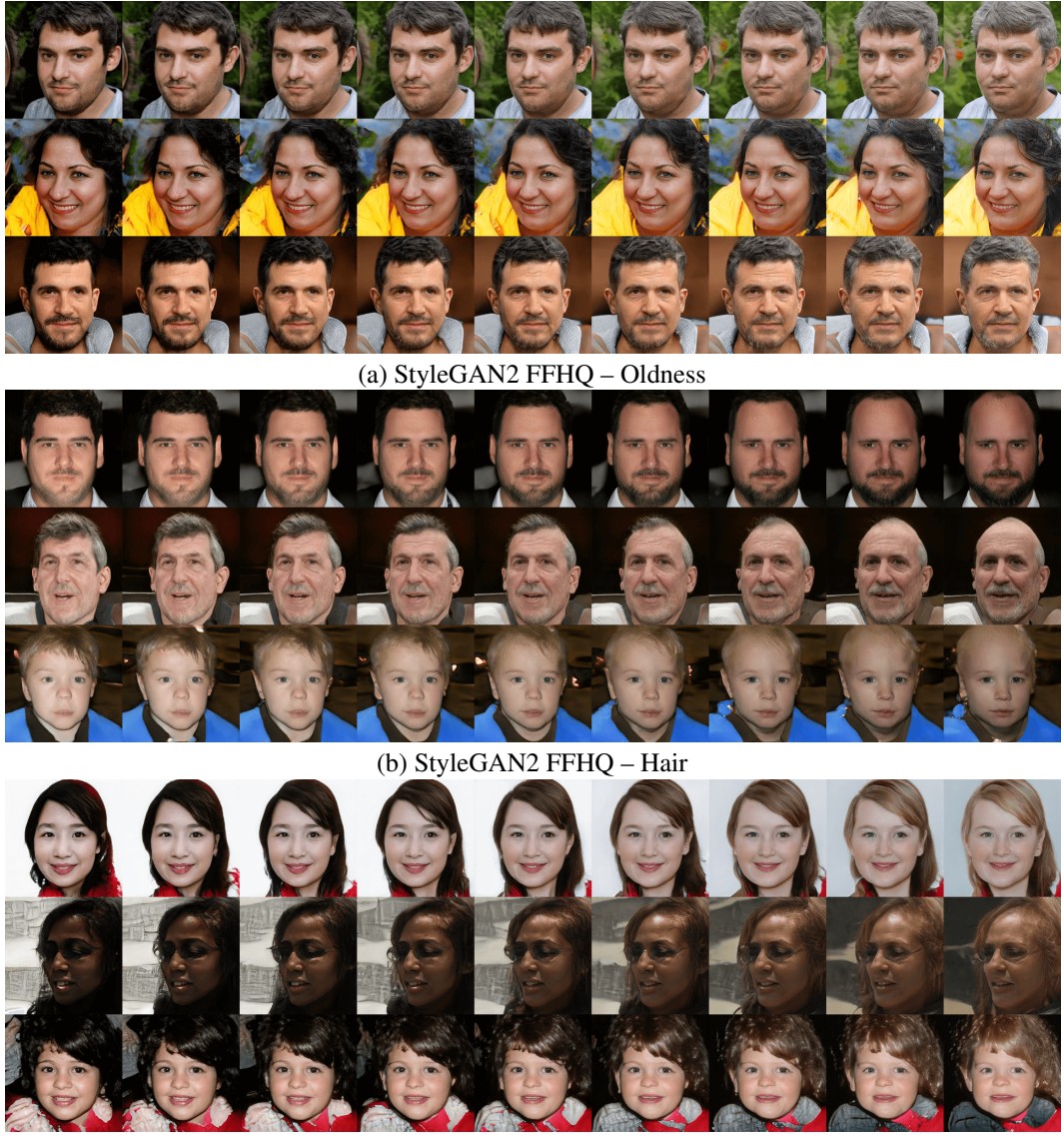

(a) StyleGAN2 FFHQ – Oldness

(b) StyleGAN2 FFHQ – Hair

(c) StyleGAN2 FFHQ – Race

Figure 19: Examples of disentangled directions for StyleGAN2 on FFHQ discovered by `DisCo`.

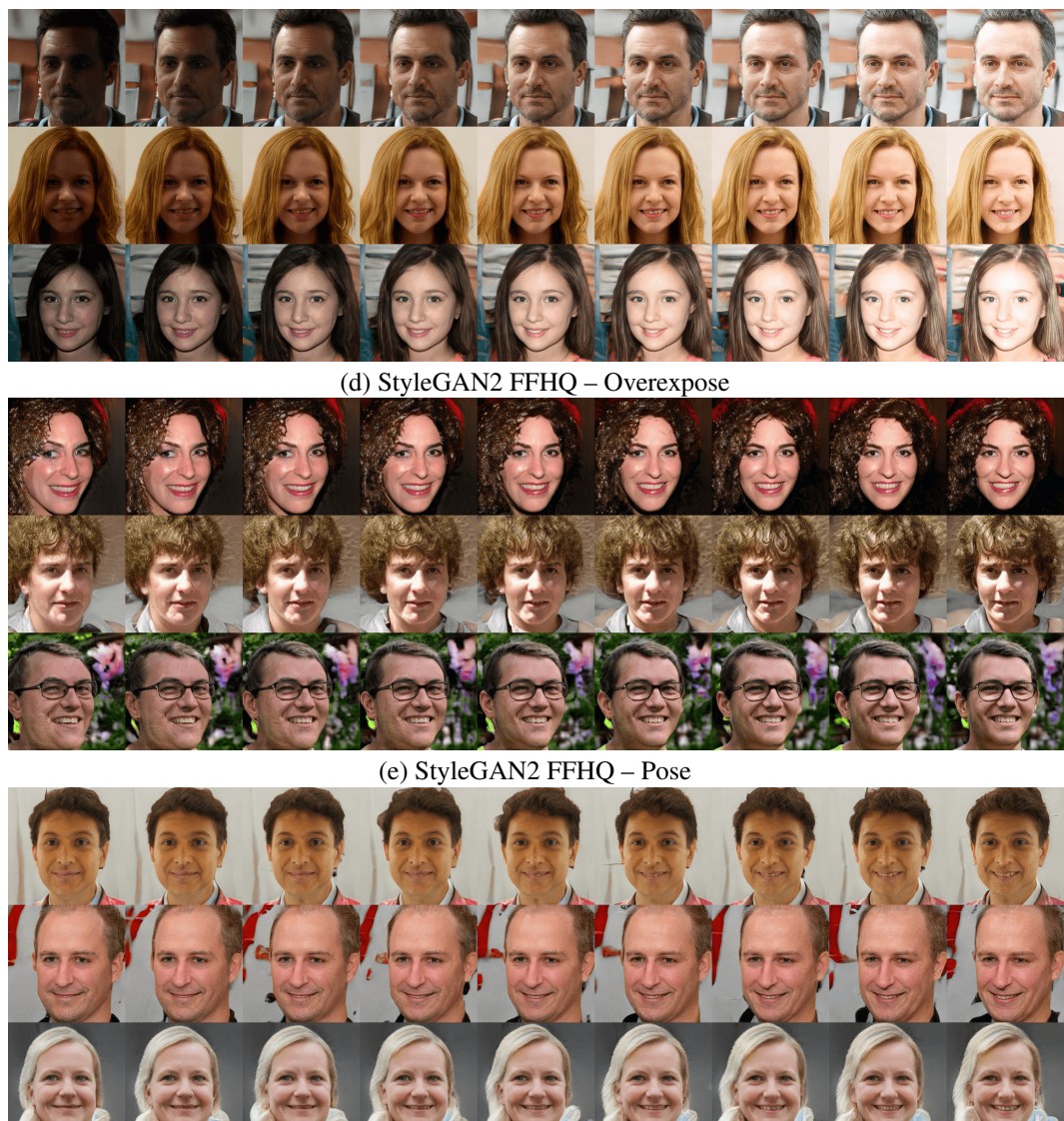

(d) StyleGAN2 FFHQ – Overexpose

(e) StyleGAN2 FFHQ – Pose

(f) StyleGAN2 FFHQ – Smile

Figure 20: Examples of disentangled directions for StyleGAN2 on FFHQ discovered by `DisCo`.

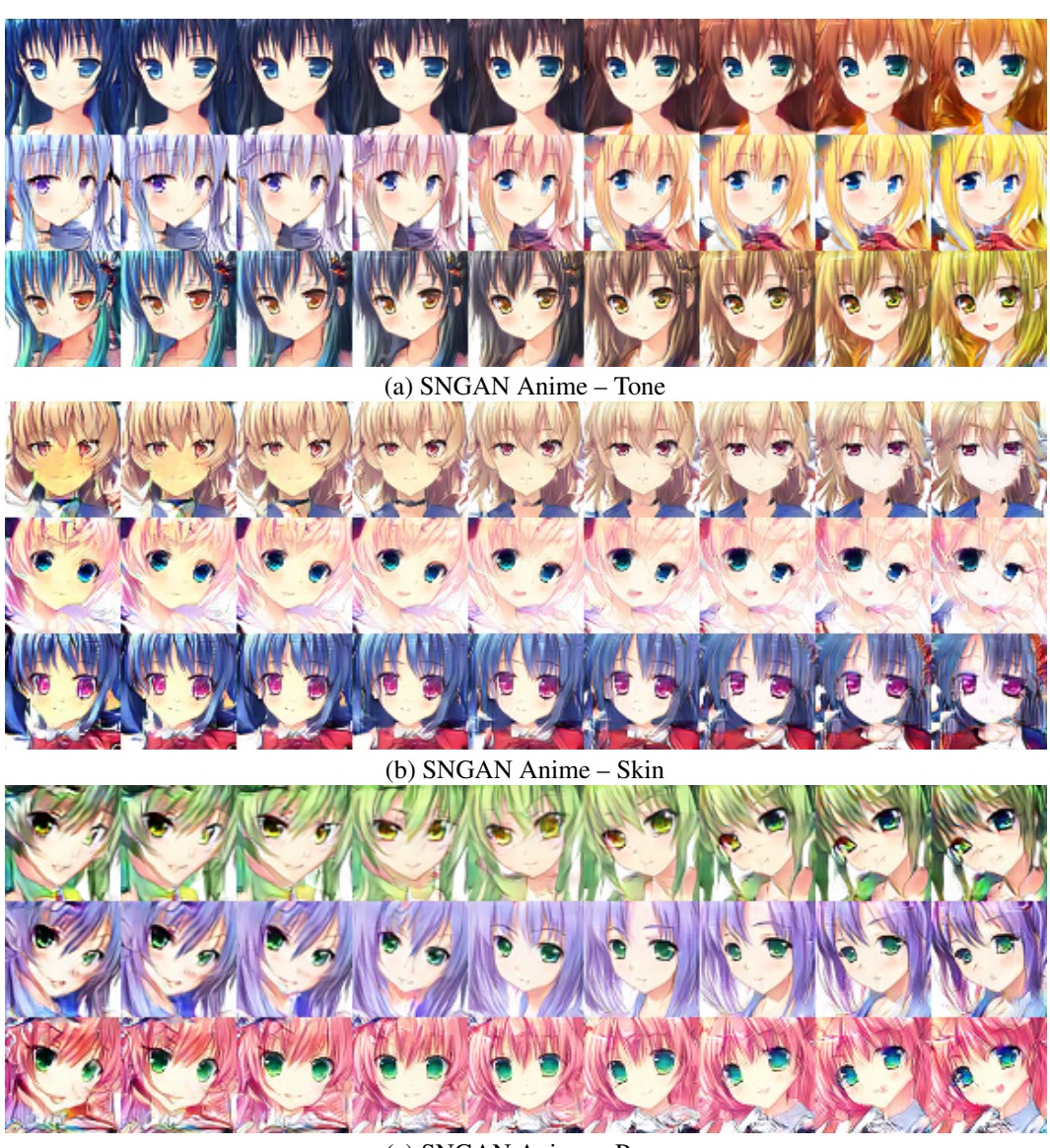

(a) SNGAN Anime – Tone

(b) SNGAN Anime – Skin

(c) SNGAN Anime – Pose

Figure 21: Examples of disentangled directions for SNGAN on Anime discoverd by `DisCo`.

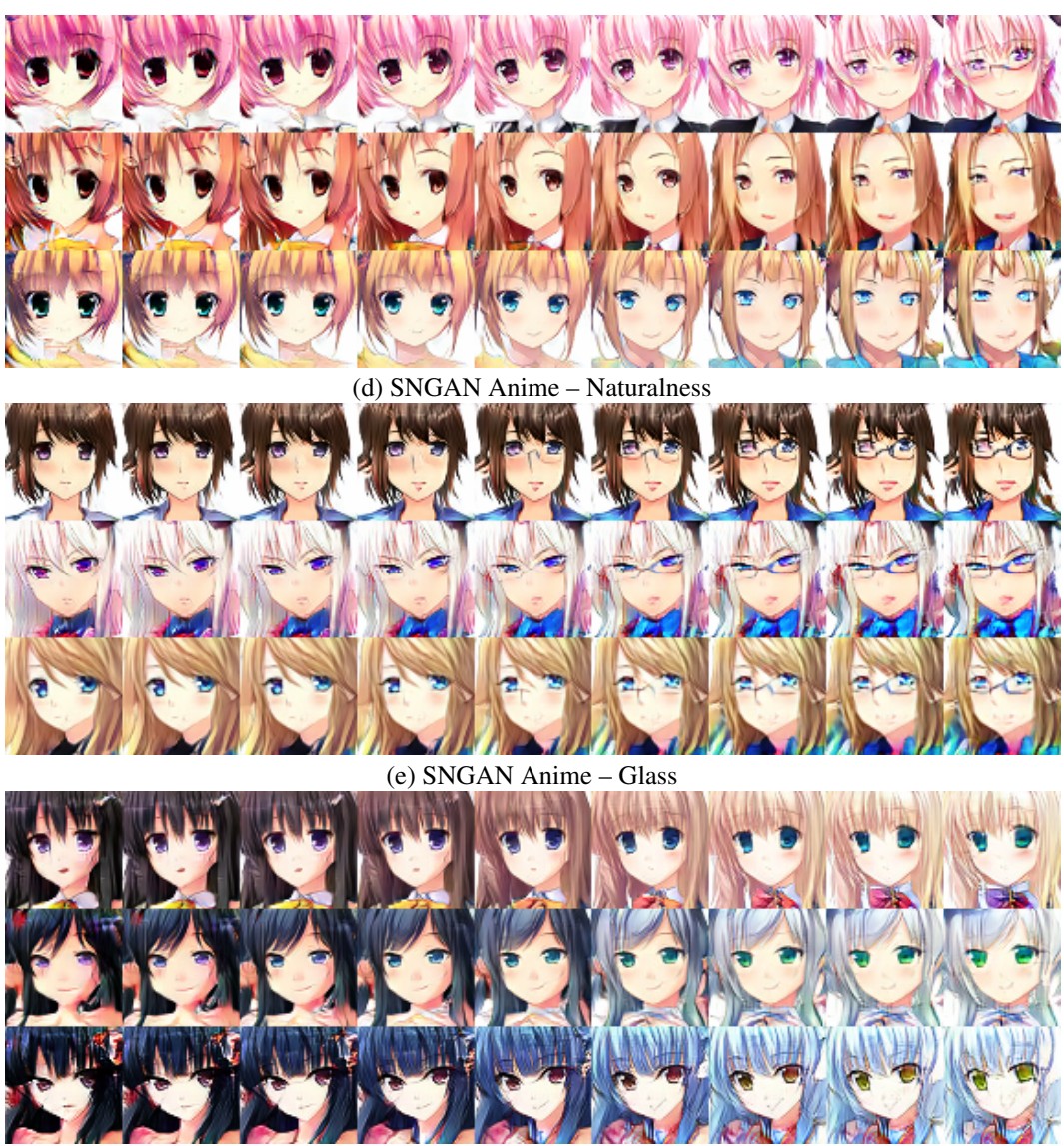

(d) SNGAN Anime – Naturalness

(e) SNGAN Anime – Glass

(f) SNGAN Anime – Whiteness

Figure 22: Examples of disentangled directions for SNGAN on Anime discovered by DisCo.

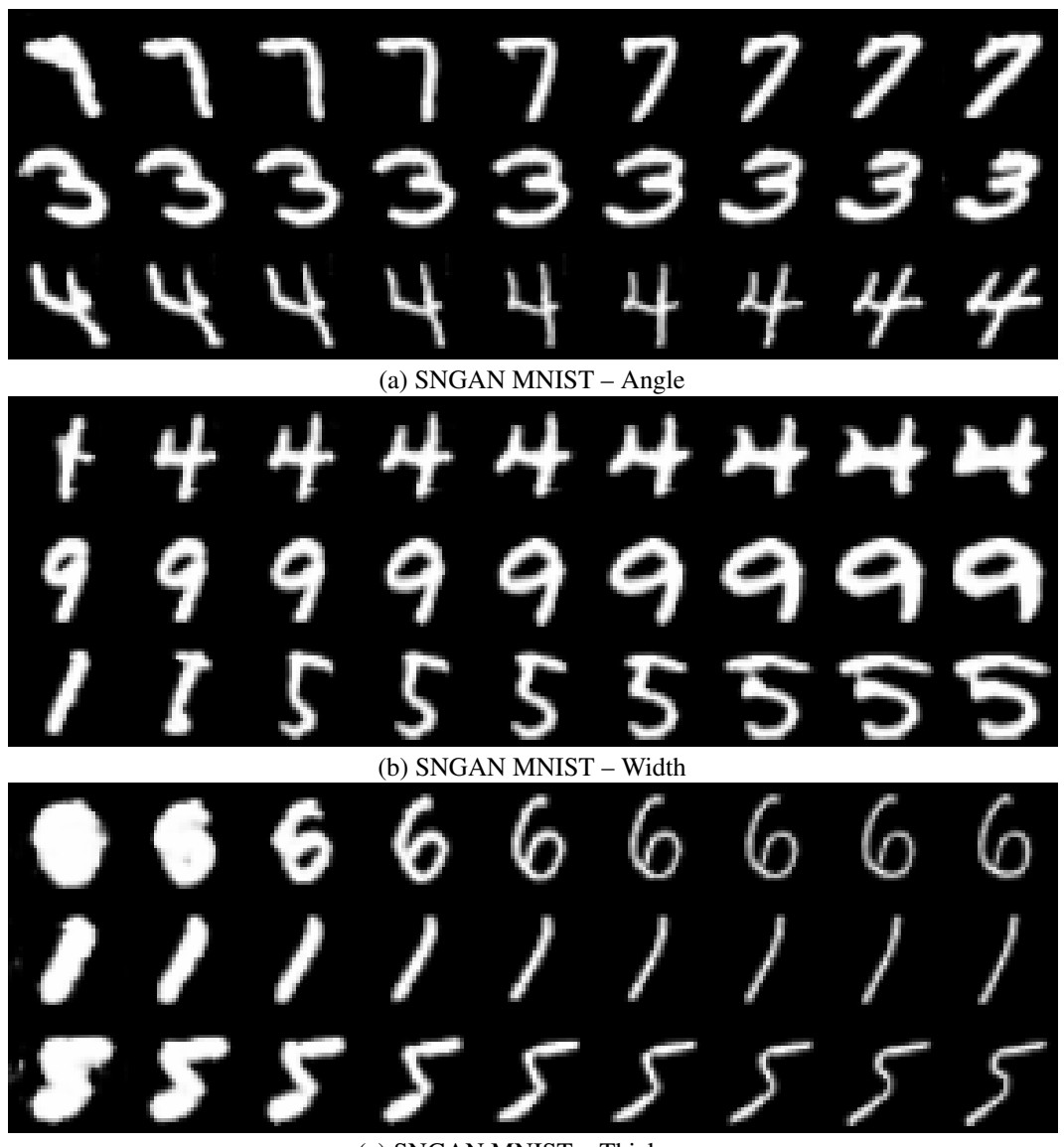

(a) SNGAN MNIST – Angle

(b) SNGAN MNIST – Width

(c) SNGAN MNIST – Thickness

Figure 23: Examples of disentangled directions for SNGAN on MNIST discovered by `DisCo`.

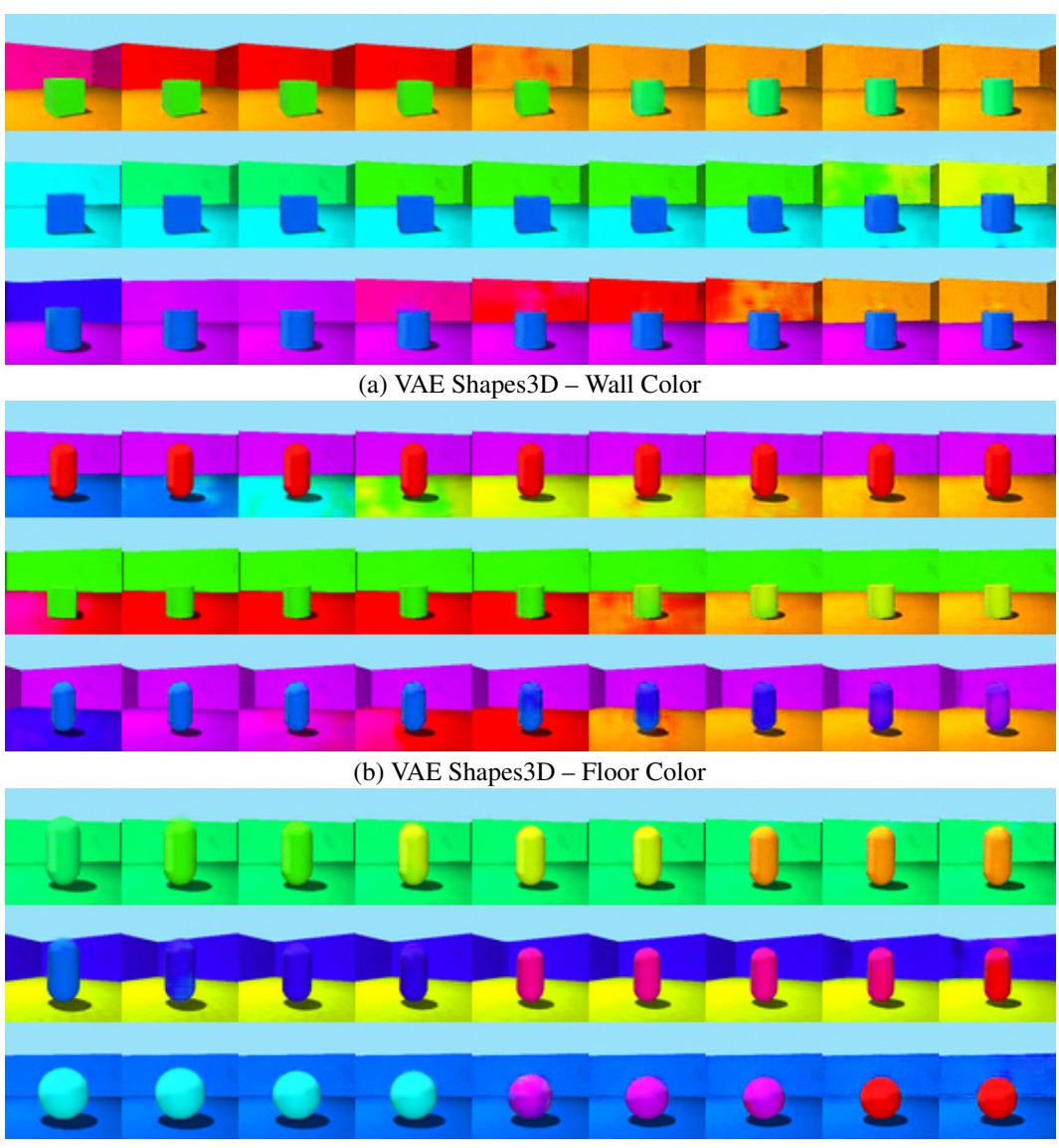

(a) VAE Shapes3D – Wall Color

(b) VAE Shapes3D – Floor Color

(c) VAE Shapes3D – Object Color

Figure 24: Examples of disentangled directions for VAE on Shapes3D discovered by `DisCo`.

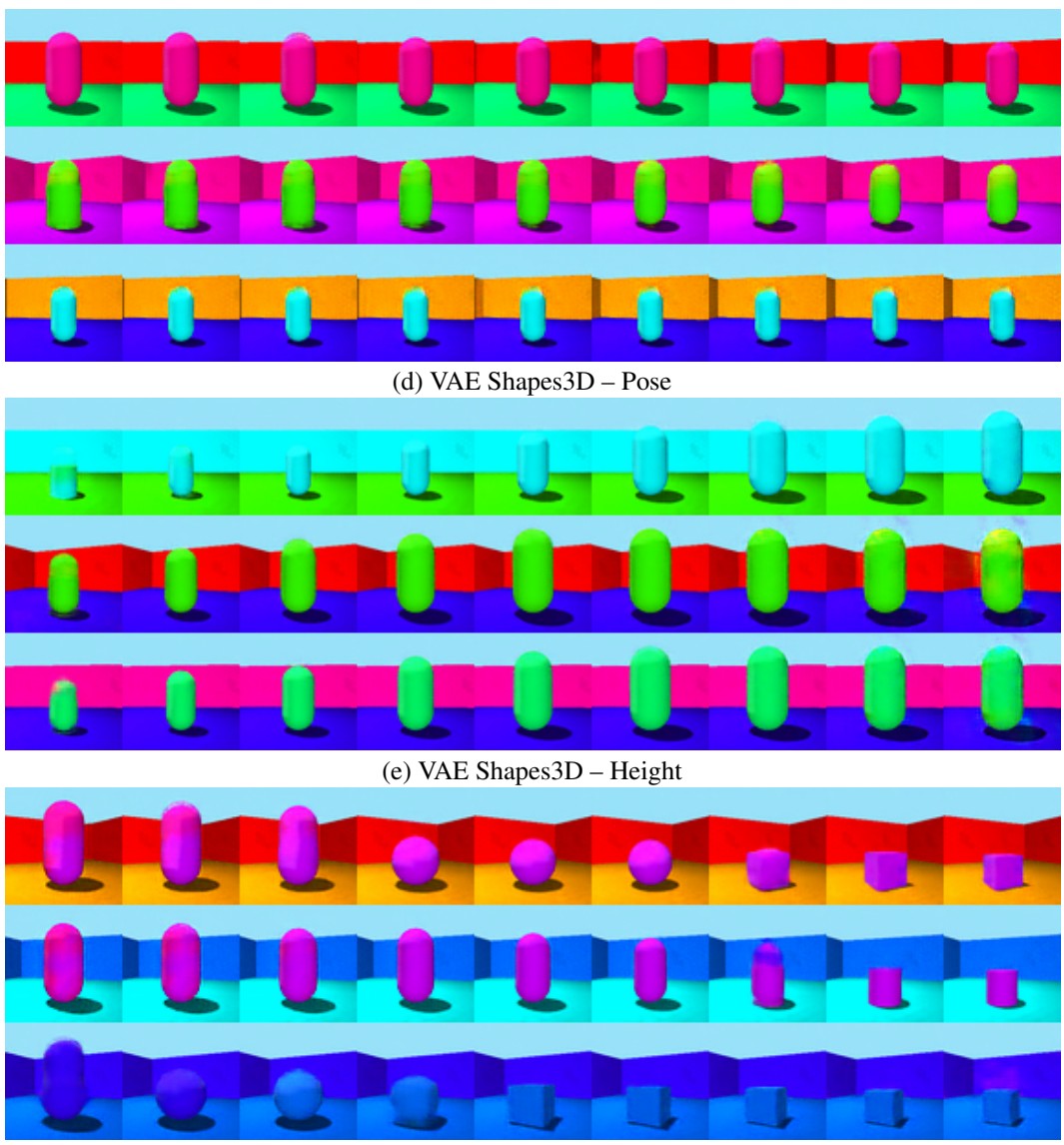

(d) VAE Shapes3D – Pose

(e) VAE Shapes3D – Height

(f) VAE Shapes3D – Object Shape

Figure 25: Examples of disentangled directions for VAE on Shapes3D discovered by `DisCo`.

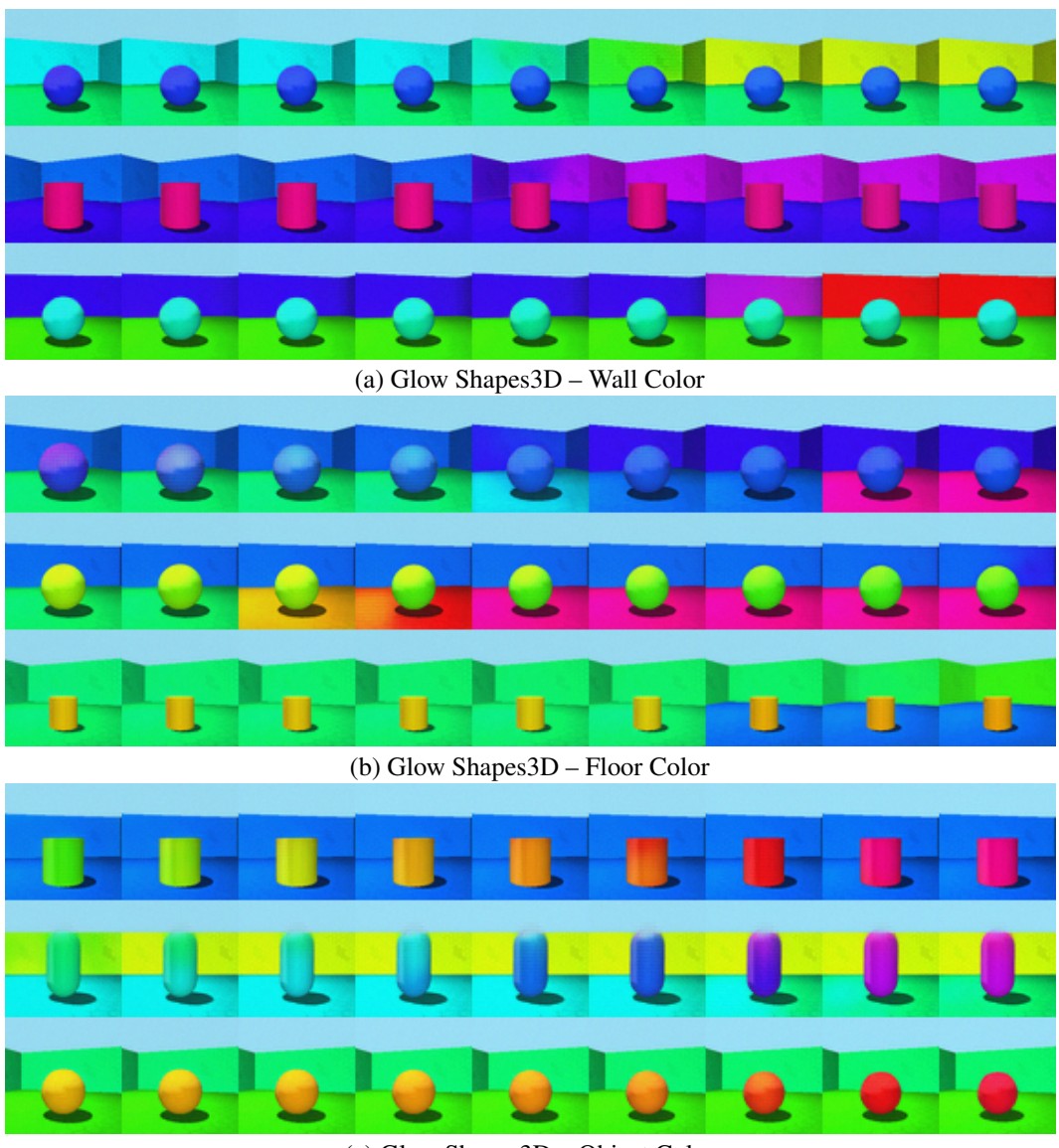

(a) Glow Shapes3D – Wall Color

(b) Glow Shapes3D – Floor Color

(c) Glow Shapes3D – Object Color

Figure 26: Examples of disentangled directions for Glow on Shapes3D discovered by `DisCo`.

## D    AN INTUITIVE ANALYSIS FOR DISCO

DisCo works by contrasting the variations resulted from traversing along the directions provided by the Navigator. Is the method sufficient to converge to the disentangled solution? Note that it is very challenging to answer this question. To our best knowledge, for unsupervised disentangled representation learning, there is no sufficient theoretical constraint to guarantee the convergence to a disentangled solution Locatello et al. (2019). Here we provide an intuitive analysis for DisCo and try to provide our thoughts on how DisCo find the disentangled direction in the latent space, which is supported by our observations on pretrained GAN both quantitatively and qualitatively. The intuitive analysis consists of two part: $(i)$ The directions that can be discovered by DisCo have different variation patterns compared to random directions. $(ii)$ DisCo hardly converges to the an entangled solution.

### D.1    WHAT KIND OF DIRECTIONS DISCO CAN CONVERGE TO?

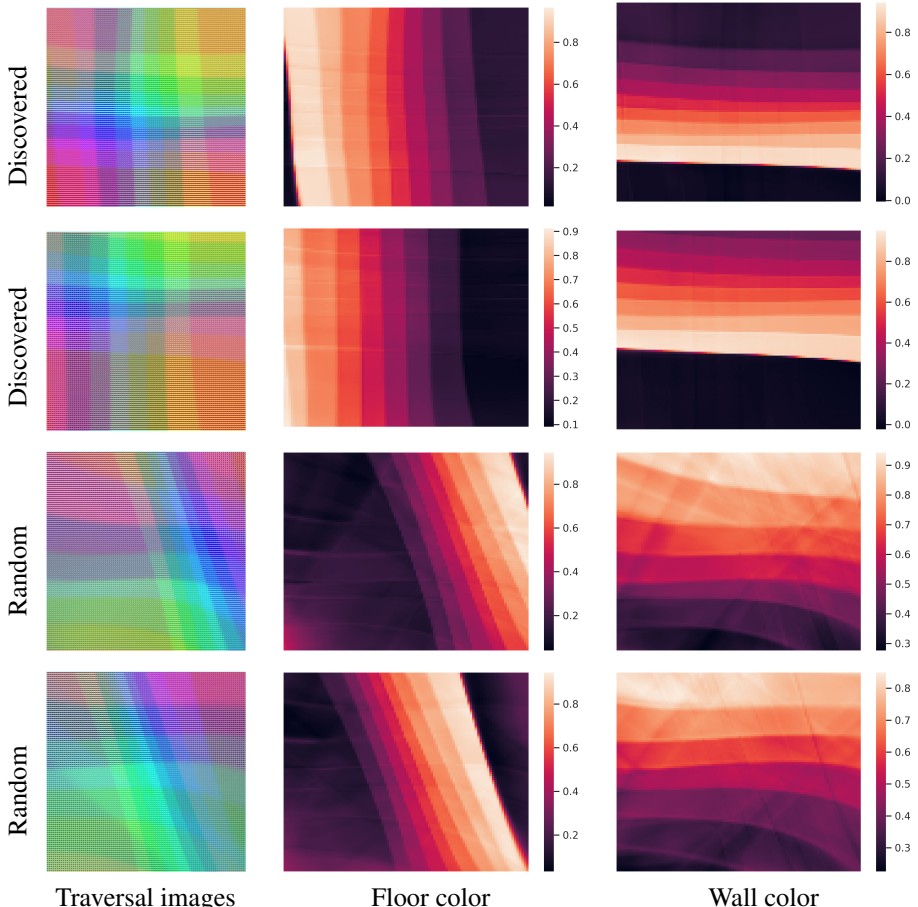

Figure 27: Visualization of the latent space of GAN on Shapes3D with discovered directions from DisCo or random sampled directions. We traverse the latent space with a range of $[-25, 25]$ and a step of $0.5$, which results in $10,000$ ($100 \times 100$) samples.

First, we visualize the latent space and show that there are some variation patterns in the latent space for disentangled factors. We design the following visualization method. Given a pretrained GAN and two directions in the latent space, we traverse along the plane expanded by the two directions to generate a grid of images. The range is large enough to include all values of these disentangled factors, and the step is small enough to obtain a dense grid. Then, we input these images into an encoder that trained with ground truth factors labels. We get a heatmap of each factor (the value is the response value corresponding dimension of the factor). In this way, we can observe the variation pattern that emerged in the latent space.

We take the pretrained StyleGAN on Shapes3D (synthetic) and FFHQ (real-world). For Shapes3D, we take background color and floor color as the two factors (since they refer to different areas in the image, these two factors are disentangled). For FFHQ, we take smile (mouth) and bald (hair) as the two factors (disentangled for referring to different areas). We then choose random directions and the directions discovered by DisCo. The results are shown in Figure 27 and Figure 28.

We find a clear difference between random directions and directions discovered by DisCo. This is because DisCo can learn the directions by separating the variations resulted from traversing along with them. However, not all directions can be separated. For those directions in which the variations are not able to be recognized or clustered by the encoder $E$, it is nearly impossible for DisCo to converge to them. Conversely, for those directions that can be easily recognized and clustered, DisCo will converge to them with a higher probability. From the following observations, we find that the variation patterns resulting from the directions corresponding to disentangled factors are easily recognized and clustered.

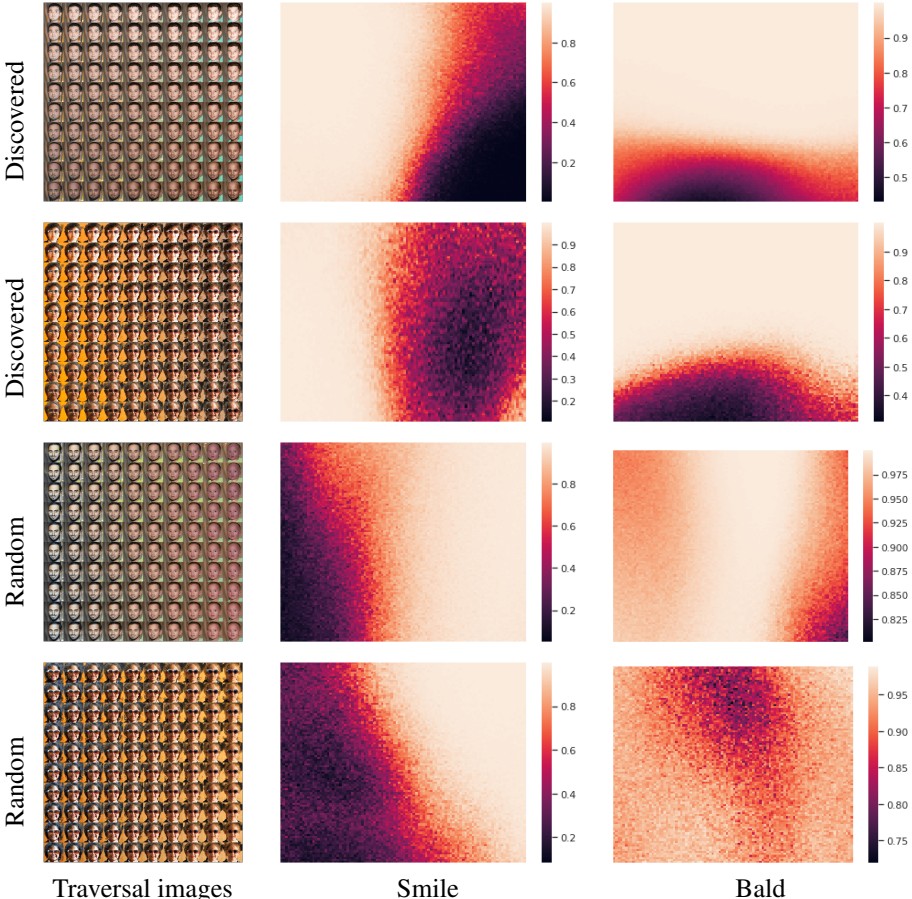

Figure 28: Visualization of the latent space of GAN on FFHQ with discovered directions from DisCo or random sampled directions. We traverse the latent space with a range of $[-15, 15]$ and a step of $0.3$, which results in $1,0000$ ($100 \times 100$) samples. For better visualization, we only present the traversal results with a step of $5$ ($10 \times 10$).

## D.2 WHY DisCo HARDLY CONVERGES TO THE ENTANGLED CASES?

In the previous section, we show that DisCo can discover the directions with distinct variation patterns and exclude random directions. Here we discuss why DisCo can hardly converge to the following entangled case (trivial solution based on disentangled one). For example, suppose there is an entangled direction of factors A and B (A and B change with the same rate when traversing along with it) in the latent space of generative models, and DisCo can separate the variations resulting

Latent Space of Generative Models

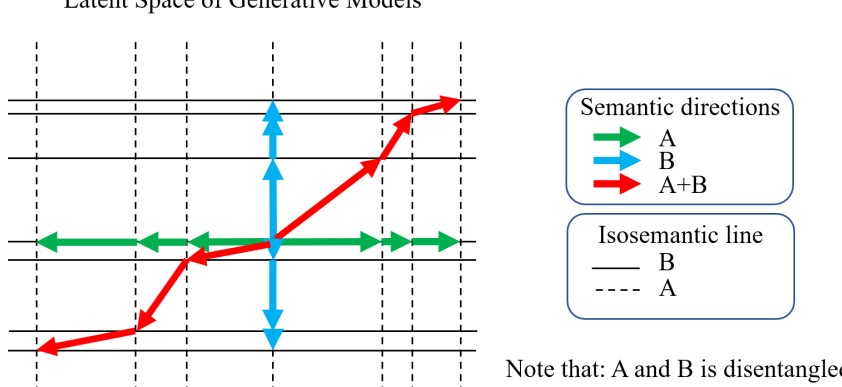

Figure 29: Sketch map of latent space of generative models.

from the direction of A and the entangled direction. In that case, `DisCo` has no additional bias to update these directions to converge to disentangled ones.

In the following text, for ease of referring to, we denote the entangled direction of factors A and B (A and B change with the same rate when traversing along with it) as A+B direction, and direction of factor A (only A change when we traverse along with it). The reasons for why `DisCo` is hardly converged to the case of A and A+B are two-fold:

$(i)$ Our encoder is a lightweight network (5 CNN layers + 3 FC layers). It is nearly impossible for it to separate the A and A+B directions.

$(ii)$ In the latent space of the pretrained generative models, the disentangled directions (A, B) are consistent at different locations. In contrast, the entangled directions (A+B) are not, as shown in Figure 29.

We conduct the following experiments to verify them. For $(i)$, we replace our encoder in `DisCo` with a ResNet-50 and train `DisCo` from scratch on the Shapes3D dataset. The loss, MIG, and DCI are presented in Table 11. The trivial solution is possible when the encoder is powerful enough to fit the A and A+B directions to "become orthogonal". With this consideration, in `DisCo` we adopt a lightweight encoder to avoid this issue.

| | Our Encoder | ResNet-50 |
|---|---|---|
| Param | 4M | 25.5M |
| Loss ($\downarrow$) | 0.550 | 0.725 |
| MIG ($\uparrow$) | 0.562 | 0.03 |
| DCI ($\uparrow$) | 0.736 | 0.07 |

Table 11: Ablation study on encoder of `DisCo`.

For $(ii)$, as the sketch Figure 29 demonstrates, the disentangled directions ("A"- blue color or "B"-green color) are consistent, which is invariant to the location in the latent space, while the entangled directions ("A+B"- red color) is not consistent on different locations. The fundamental reason is that: the directions of the disentangled variations are invariant with the position in the latent space. However, the "rate" of the variation is not. E.g., at any point in the latent space, going "up" constantly changes the camera's pose. However, at point a, going "up" with step 1 means rotating 10 degrees. At point b, going "up" with step 1 means rotating 5 degrees. When the variation "rate" of "A" and "B" are different, the "A+B" directions at different locations are not consistent.

Based on the different properties of disentangled and entangled directions in the latent space, `DisCo` can discover the disentangled directions with contrastive loss. The contrastive loss can be understood from the clustered view (Wang & Isola, 2020; Li et al., 2021b). The variations from the disentangled directions are more consistent and can be better clustered compared to the variations from the

entangled ones. Thus, `DisCo` can discover the disentangled directions in the latent space and learn disentangled representations from images. We further provide the following experiments to support our above analysis.

### D.2.1 QUANTITATIVE EXPERIMENT

We compare the losses of three different settings:

- $A$: For a navigator with disentangled directions, we fix the navigator and train the encoder until convergence.
- $A + B$: For a navigator with entangled directions (we use the linear combination of the disentangled directions to initialize the navigator), we fix it and train the encoder until convergence.
- $A + B \rightarrow A$: After A+B is convergent, we update both the encoder and the navigator until convergence.

The Contrastive loss after convergence is presented in Table 12.

|      | $A$    | $A + B$ | $A + B \rightarrow A$ |
|------|--------|---------|-----------------------|
| Loss | 0.5501 | 0.7252  | 0.5264                |

Table 12: Loss comparison on different settings.

The results show that: $(i)$ The disentangled directions (A) can lead to lower loss and better performance than entangled directions (A+B), indicating no trivial solution. $(ii)$ Even though the encoder with A+B is converged, when we optimize the navigator, gradients will still backpropagate to the navigator and converge to A.

### D.2.2 QUALITATIVE EXPERIMENT

We visualize the latent space of GAN in Figure 30 to verify the variation "rate" in the following way: in the latent space, we select two ground truth disentangled directions: floor color (A) and background color (B) obtained by supervision with InterFaceGAN (Shen et al., 2020), we conduct equally spaced sampling along the two disentangled directions: A (labeled with green color variation), B (labeled with gradient blue color) and composite direction (A+B, labeled with gradient red color) as shown in Figure 30 (a).

Then we generate the images (also include other images on the grid as shown in Figure 30 (b) ), and feed the images in the bounding boxes into a "ground truth" encoder (trained with ground truth disentangled factors) to regress the "ground truth" disentangled representations of the images.

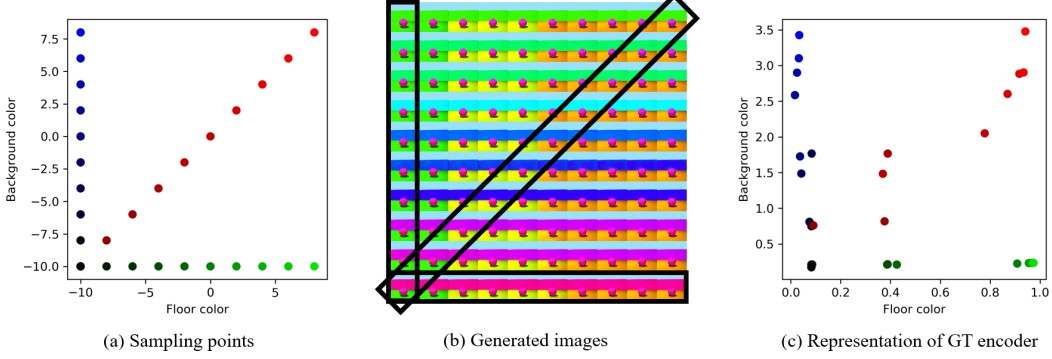

(a) Sampling points      (b) Generated images      (c) Representation of GT encoder

Figure 30: Visualization of GAN latent space.

In Figure 30 (c), the points labeled with green color are well aligned with the x-axis indicating only floor color change, points labeled with blue variation are well aligned with the y-axis indicating only

background color change. However, the points labeled with red color are NOT aligned with any line, which indicates the directions of A+B are not consistent. Further, the variation "rate" is relevant to the latent space locations for the two disentangled directions. This observation well supports our idea shown in Figure 29. The different properties between disentangled and entangled directions enable `DisCo` to discover the disentangled directions in the latent space.

## E  EXTENSION: BRIDGE THE PRETRAINED VAE AND PRETRAINED GAN

Researchers are recently interested in improving image quality given the disentangled representation generated by typical disentanglement methods. Lee et al.(Lee et al., 2020) propose a post-processing stage using a GAN based on disentangled representations learned by VAE-based disentanglement models. This method scarifies a little generation ability due to an additional constraint. Similarly, Srivastava et al. (Srivastava et al., 2020) propose to use a deep generative model with AdaIN (Huang & Belongie, 2017) as a post-processing stage to improve the reconstruction ability. Following this setting, we can replace the encoder in `DisCo` (GAN) with an encoder pretrained by VAE-based disentangled baselines. In this way, we can bridge the pretrained disentangled VAE and pretrained GAN, as shown in Figure 31. Compared to previous methods, our method can fully utilize the state-of-the-art GAN and the state-of-the-art VAE-based method and does not need to train a deep generative model from scratch.

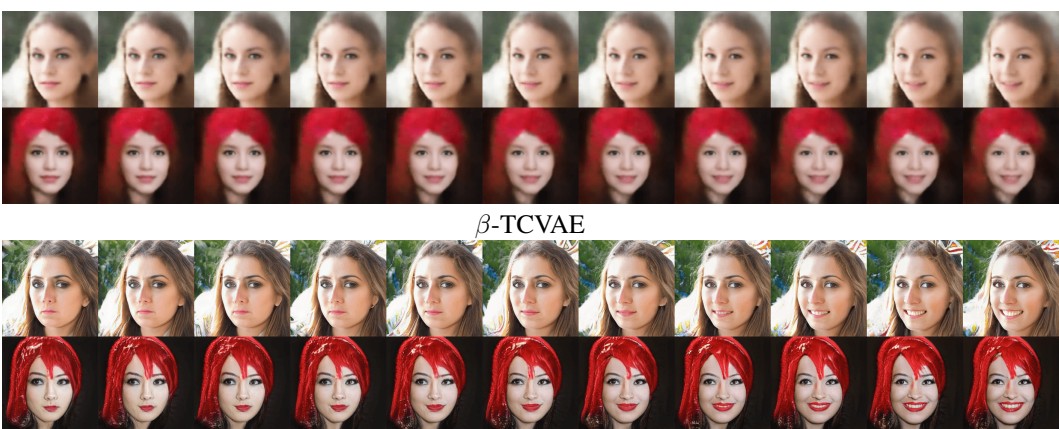

$\beta$-TCVAE

`DisCo` (with a pretrained encoder)

Figure 31: `DisCo` with a pretrained encoder allows synthesizing high-quality images by bridging pretrained $\beta$-TCVAE and pretrained StyleGAN2.

## F    DISCUSSION ON RELATION BETWEEN BCELOSS AND NCELOSS

We would like to present a deep discussion on the relation between the BCELoss $\mathcal{L}_{logits}$ and NCELoss $\mathcal{L}_{NCE}$. This discussion is related to the NCE paper Gutmann & Hyvärinen (2010), and InfoNCE paper van den Oord et al. (2018). The discussion is as following: $(i)$ we first provide a formulation of a general problem and get two objectives, $\mathcal{L}_1$ and $\mathcal{L}_2$, and $\mathcal{L}_1$ is the upper bound of $\mathcal{L}_2$. $(ii)$ Following Gutmann & Hyvärinen (2010), we show that $\mathcal{L}_1$ is aligned with $\mathcal{L}_{BCE}$ under the setting of Gutmann & Hyvärinen (2010). $(iii)$ Following van den Oord et al. (2018), we prove $\mathcal{L}_2$ is aligned with $\mathcal{L}_{NCE}$ under the setting of van den Oord et al. (2018). $(iii)$ We discuss the relation between these objectives and the loss in our paper.

**Part I.**    Assume we have $S$ observations $\{x_i\}_{i=1}^S$ from a data distribution $p(x)$, each with a label $C_i \in \{0, 1\}$. The we denote the posterior probabilities as $p^+(x) = p(x|C = 1)$ and $p^-(x) = p(x|C = 0)$.

We define two objectives as follow:

$$\mathcal{L}_1 = -\sum_{i=1}^S C_i \log P(C_i = 1|x_i) + (1 - C_i) \log P(C_i = 0|x_i), \tag{10}$$

and

$$\mathcal{L}_2 = -\sum_{i=1}^S C_i \log P(C_i = 1|x_i) \tag{11}$$

Since $-\sum_{i=1}^S (1 - C_i) \log p(C_i = 0|x_i) \geq 0$, we have

$$\mathcal{L}_1 \geq \mathcal{L}_2. \tag{12}$$

$\mathcal{L}_1$ is the upper bound of $\mathcal{L}_2$.

This a general formulation of a binary classification problem. In the context of our paper, we have a paired observation $x_i : (q, k_i)$, with $q$ as the query, and the key $k_i$ is either from a positive key set $\{k_j^+\}_{j=1}^N$ or as negative key set $\{k_m^-\}_{m=1}^M$ (i.e., $\{k_i\}_{i=1}^{N+M} = \{k_j^+\}_{j=1}^N \bigcup \{k_m^-\}_{m=1}^M$), where $M = S - N$. And $C_i$ is assigned as:

$$C_i = \begin{cases} 1, & k_i \in \{k_j^+\}_{j=1}^N \\ 0, & k_i \in \{k_m^-\}_{m=1}^M \end{cases} \tag{13}$$

In our paper, we have $h(x) = \exp(q \cdot k/\tau)$.

**Part II.**    In this part, following Gutmann & Hyvärinen (2010), we show that $\mathcal{L}_1$ is aligned with $\mathcal{L}_{logits}$ (Equation 3 in the main paper) under the setting of Gutmann & Hyvärinen (2010). Following Gutmann & Hyvärinen (2010)), we assume the prior distribution $P(C = 0) = P(C = 1) = 1/2$, according to the Bayes rule, we have

$$P(C = 1|x) = \frac{p(x|C = 1)P(C = 1)}{p(x|C = 1)P(C = 1) + p(x|C = 0)P(C = 0)} = \frac{1}{1 + \frac{p^-(x)}{p^+(x)}}. \tag{14}$$

And $P(C = 0|x) = 1 - P(C = 1|x)$.

On the other hand, we have a general form of BCELoss, as

$$\mathcal{L}_{BCE} = -\sum_{i=1}^S C_i \log \sigma(q \cdot k_i/\tau) + (1 - C_i) \log(1 - \sigma(q \cdot k_i/\tau)), \tag{15}$$

where $\sigma(\cdot)$ is the sigmoid function. We have

$$\sigma(q \cdot k/\tau) = \frac{1}{1 + \exp(-q \cdot k/\tau)} = \frac{1}{1 + \frac{1}{\exp(q \cdot k/\tau)}} = \frac{1}{1 + \frac{1}{h(x)}}, \tag{16}$$

From Gutmann & Hyvärinen (2010) Theorem 1, we know that when $\mathcal{L}_{BCE}$ is minimized, we have

$$h(x) = \frac{p^+(x)}{p^-(x)}. \tag{17}$$

Thus, we know the BCELoss $\mathcal{L}_{BCE}$ is a approximation of the objective $\mathcal{L}_1$.

**Part. III**  Following van den Oord et al. (2018), we prove $\mathcal{L}_2$ is aligned with $\mathcal{L}_{NCE}$ (Equation 2 in the main paper) under the setting of van den Oord et al. (2018)

From the typical contrastive setting (one positive sample, others are negative samples, following van den Oord et al. (2018)), we assume there is only one positive sample, others are negatives in $\{x_i\}_{i=1}^S$. Then, the probability of $x_i$ sample from $p^+(x)$ rather then $p^-(x)$ is as follows,

$$P(C_i = 1 | x_i) = \frac{p^+(x_i)\Pi_{l \neq i} p^-(x_l)}{\sum_{j=1}^S p^+(x_j)\Pi_{l \neq i} p^-(x_l)} = \frac{\frac{p^+(x_i)}{p^-(x_i)}}{\sum_{j=1}^S \frac{p^+(x_j)}{p^-(x_j)}} \tag{18}$$

From van den Oord et al. (2018), we know that when minimize Equation 11, we have $h(x) = \exp(q \cdot k/\tau) \propto \frac{p_+(x)}{p_-(x)}$. In this case, we get the form of $\mathcal{L}_{NCE}$ as

$$\mathcal{L}_{NCE} = -\sum_{i=1}^S C_i \log \frac{\exp(q \cdot k_i/\tau)}{\sum_{j=1}^S \exp(q \cdot k_j/\tau)} \tag{19}$$

$\mathcal{L}_{NCE}$ is a approximate of $\mathcal{L}_2$.

**Part. IV**  When generalize the contrastive loss into our setting ($N$ positive samples, $M$ negative samples). The BCELoss (Equation 15) can be reformulated as

The BCELoss (Equation 15) can be reformulated as

$$\hat{\mathcal{L}}_{BCE} = -\sum_{j=1}^N \log \sigma(q \cdot k_j^+/\tau) - \sum_{m=1}^M \log(1 - \sigma(q \cdot k_m^-/\tau)). \tag{20}$$

Similarly, the NCEloss (Equation 19) can be reformulated as

$$\hat{\mathcal{L}}_{NCE} = -\sum_{j=1}^N \log \frac{\exp(q \cdot k_j^+/\tau)}{\sum_{s=1}^{M+N} \exp(q \cdot k_s/\tau)} \tag{21}$$

$\hat{\mathcal{L}}_{BCE}$ is aligned with $\mathcal{L}_{logits}$ (Equation 3 in our main paper), and $\hat{\mathcal{L}}_{NCE}$ is aligned with $\mathcal{L}_{NCE}$ (Equation 2 in the main paper).

Now we have $\mathcal{L}_1$ (approximated by $\mathcal{L}_{BCE}$) is the upper bound of $\mathcal{L}_2$ (approximated by $\mathcal{L}_{NCE}$). However, as you may notice, the assumptions we made in **Part II** and **Part III** are different, one is $P(C = 0) = P(C = 1)$, the other one is only one positive sample, others are negative. Also the extent to our situation is more general case ($N$ positives, others are negatives).

However, they have the same objective, which is by contrasting positives and negatives, we can use $h(x) = exp(q \cdot k/\tau)$ to estimate $p^+/p^-$. We can think the $h(x)$ as a similarity score, i.e. if $q$ and $k$ are from a positive pair (they have the same *direction* in our paper), $h(x)$ should be as large as possible ($p^+/p^- > 1$) and vice versa. From this way, we can learn the representations $(q, k)$ to reflect the image variation, i.e., similar variations have higher score $h(x)$, while different kinds of variation have lower score $h(x)$. Then with this meaningful representation, in the latent space, can help to discover the directions carrying different kinds of image variation. This is an understanding, from a contrastive learning view, of how our method works.

