# OpenReview forum: "Learning Disentangled Representation by Exploiting Pretrained Generative Models:  A Contrastive Learning View"
_ICLR.cc/2022/Conference — ICLR 2022 Poster_

### Official Review · Reviewer_3Z9R · 2021-10-30

**Correctness:** 4
**Technical Novelty And Significance:** 3
**Empirical Novelty And Significance:** 3
**Recommendation:** 8
**Confidence:** 5

**Main Review:**

Strengths:
1. Learning disentangled representations via contrastive learning on pretrained models is an interesting direction for this community.
2. The introduced method is intuitive, simple, and effective.
3. Extensive experiments on disentanglement learning and high-resolution datasets are conducted with satisfying results obtained.
4. The introduced entropy-based domain loss and the hard negative flipping technique are effective and practical.
5. The proposed method is general for multiple generative models.

Problem:
For qualitative results on StyleGAN2 (Fig. 4 and Fig. 17 - 20), I wonder is there any manual selection on the layers to modify in the w space (like done in the methods GS and CF) or the modification happens globally on all layers in the w space?

**Summary Of The Paper:**

This paper proposes to learn disentangled representations via contrastive learning on well-pretrained generative models. Extensive experiments are conducted on various datasets and the results validate the effectiveness of the method.

**Summary Of The Review:**

The introduced method is simple, effective, and general for disentanglement learning. I recommend accept for this paper.

---

> ### Author Response · Authors · 2021-11-21
> **Response to Reviewer 3Z9R**
>
> Thanks for reviewing our paper, we appreciate the positive feedback for our work.
>
> Our modification happens globally on all layers in the W space without any manual selection. We have clarified this in the modified version in Appendix A.

---

### Official Review · Reviewer_Go6R · 2021-11-01

**Correctness:** 4
**Technical Novelty And Significance:** 3
**Empirical Novelty And Significance:** 4
**Recommendation:** 8
**Confidence:** 4

**Main Review:**

Good:

The idea is very simple and easy to implement.

The paper is very well written and easy to understand.

There are extensive ablations that show the effect of design choices and hyper-parameters.

The qualitative results look very good for disentangling, the proposed method preserves e.g. the identity much better when changing other attributes, like smile or baldness.

Quantitatively the proposed method shows better performance than the baseline for many datasets and 3 different kinds of generative model: GAN, VAE and Flow. This is impressive and shows the methods generality.

Bad:

Although the paper explained the method well from the perspective of reproducibility, it does not explain why the method should chose semantically meaningful directions. One can imagine a shortcut scenario, where the method learn 0.5a+0.5b and 0.5a-0.5b directions, where a and b are perfect semantically meaningful directions. In principle the training loss could be minimised with this solution as well (?). The reason why this does not happens is because of the implicit biases in the networks (?). But then why is this method performing better than prior works?

"(ii) the factors are embedded in the pretrained model, severing as an inductive bias for unsupervised disentangled representation learning."
This still allows for the mixed solution 0.5a+0.5b and 0.5a-0.5b.

I think the following statement is incorrect, it should be removed:
"A composed of 3 fully-connected layers performs poorly, indicating the disentangled directions of the latent space W of StyleGAN is nearly linear."
- W is nearly linear because there are good directions in it, and a linear method can perform well in it.
- The 3 layer network fails for some other reason, in principle it should work at least as good as the linear model, as it has the preresentation capacity.

The method is very sensitive to the ratio between positive and negative samples. A very good tuning is needed, which is shown in the paper for most hyper-parameters. One might think that the gains come from the extensive tuning rather than the proposed idea itself.

minor:
- Although the images are resized to 64x64, it would be nice to see full resolution results with e.g. the StyleGAN2 generator. Or was the generator also retrained with reduced size images (for faster training I guess)?
- some typos and grammar could be fixed, e.g. "... generative model are two-ford: ..."


**Summary Of The Paper:**

The paper proposes a novel representation learning technique to disentangle the latent space of pre-trained generative models, by discovering semantically meaningful directions in them.

The method trains a navigator and a delta-contrastor network, which consists of 2 encoders sharing weights. First, random samples are perturbed along the directions obtained from the navigator. The perturbed vectors are then decoded with the pre-trained generator, then encoded and the difference between 2 samples are taken. The output is in the variation space, where a contrastive learning technique clusters together the samples that were perturbed with the same direction.


**Summary Of The Review:**

I think the paper contains good practical ideas and a very extensive experimental evaluation.

The main weakness of the paper is the lack of deep insights. The reader learns a simple idea that is easy to implement and works well in practice, but does not get an answer to the "why" question.

Overall the good outweighs the bad.

--------- UPDATE ---------

I have read the other reviews and the rebuttals. The authors have clarified some details in the experiments. I think the experiments are strong and give valuable data for future research. I raise my score to accept.

---

> ### Author Response · Authors · 2021-11-21
> **Response to Reviewer Go6R**
>
> Thanks for reviewing our paper, and we appreciate the positive feedback for our work and insightful questions.
>
> - Why does the situation of  0.5a+0.5b and 0.5a-0.5b not happen?
>
> Thank you for raising the in-depth question about “why the method should choose semantically meaningful directions”. We believe that our analysis may provide a better understanding of this problem. We think this is an interesting problem that has good discussion potentials for the community.
>
> We summarize the experiments result and our analysis here (details in Appendix D):
>
> (i) The loss of a fixed navigator (0.5a+0.5b and 0.5a-0.5b) is higher than the one of a fixed navigator (a and b).  The model will still converge to the case of a and b, even initialized with 0.5a+0.5b and 0.5a-0.5b. This indicates that the case of 0.5a+0.5b or 0.5a-0.5b is not a local minimum for DisCo.
>
> (ii) Visualization of the latent space of pretrained generative models. The semantic meaningful directions (a or b) have dominant patterns, which can not be observed for 0.5a + 0.5b or 0.5a - 0.5b.
>
> Intuitively,  based on the quantitative and qualitative results, compared to “pure variations” (with dominate patterns) from the perfect semantically meaningful directions (a and b),  the “mixed variations” (without dominate patterns) from 0.5a+0.5b or 0.5a-0.5b  are more diverse and it is harder for DisCo to pull them together.
>
> - Why is this method performing better than prior works?
>
> We would like to address your concern from the following two perspectives:
>
> (i) Why DisCo outperforms those methods based on pretrained GANs (e.g., DS).
>
> Those baselines directly compute (DS) or learn (LD)  the semantic directions first and then use an encoder to learn the disentangled representation in an extra second stage. However, in DisCo, both the encoder for disentangled representations and the Navigator for semantic directions are learned simultaneously. We argue that the learning of these two, together, can benefit each other.
>
> (ii) Why DisCo outperforms those VAE-based methods.
>
> As we can see from Fig. 1 (a), for the VAE-based methods, they have to learn both the reconstruction and disentangled representation. It has been pointed out that these methods usually suffer a trade-off between disentanglement and generation quality. However, DisCo tackles this trade-off by the proposed paradigm, as shown in Fig. 1 (b).
>
> - The incorrect statement should be removed.
>
> Thanks for pointing out this. We agree with you and have fixed this in the rebuttal version.
>
> - Hyper-parameter tuning & the ratio between positive and negative samples.
>
> Thanks for pointing this out. We would like to highlight two points on the hyper-parameters.
>
> (i) Our hyper-parameters are shared across different models and datasets (as shown in Appendix A). We don’t need to tune the hyper-parameters for any specific models on those datasets.
>
> (ii) The hyperparameter that is sensitive to the performance has been studied in our paper (the ratio of positive and negative samples).
>
> In addition, since our method does NOT need extra efforts to tune the hyperparameters (it works well on the default setting), it is easy to reproduce our results, and we also provide our code (https://bit.ly/3innjnr).
>
> [minor points]
>
> - Other resolution results with the StyleGAN2 generator.
>
> The original resolution is 64x64 for the disentangled datasets, including Shape3D, Car3D. For MPI, we resize it to 64x64 for faster training. The resolution we use is 256x256 for FFHQ, LSUN cat, LSUN church. We have clarified this in the rebuttal version.
>
> - Some typos and grammar could be fixed.
>
> Thank you for pointing this out. We have fixed this in the rebuttal version.

---

### Official Review · Reviewer_sBQs · 2021-11-02

**Correctness:** 3
**Technical Novelty And Significance:** 3
**Empirical Novelty And Significance:** 3
**Recommendation:** 6
**Confidence:** 4

**Main Review:**

Pros:
1) The proposed method is novel and achieves SOTA results in disentanglement while ensuring good generation quality.
2) Extensive experiments and ablation studies.
3) In general, the paper is well written and easy to read.

Cons:
1) There are still some flaws in the proposed method.
2) Some details about how to compute MIG and DCI for discovering-based methods are missing.
3) MIG and DCI metrics are out-of-date and may not well characterize disentanglement.


**Summary Of The Paper:**

This paper proposes DisCo, a framework that learns disentangled representations from pretrained entangled generative models. Extensive experimental results show that DisCo outperforms many baselines in both quantitative and qualitative evaluations.

**Summary Of The Review:**

1) Novelty:
- The authors propose a novel and interesting idea of learning disentangled representations by reusing the decoder/generator of pretrained generative models and only learning the directions that lead to disentanglement.

2) Correctness of the method:
- Is there an absolute operator in Eq. 1?
- $\mathcal{V} \in \mathbb{R}^{J}$ is not mathematically correct, it should be $\mathcal{V} \subset \mathbb{R}^J_+$ since $v$ is always >= 0.
- Why are the sizes of the query key set and the positive key set different? They should be the same in contrastive learning. If the authors use multiple positive keys per query key, the size of the positive key set should be $B \times N$ instead of $N$.
- I think the formula of NCE in Eq. 2 is incorrect. The sum in the nominator should be outside the $\log$ and should be treated as the mean over $N$. Which equation in (van de Oord et al., 2018) did the authors refer to?
- I would like to see mathematical proof of why the BCELoss $\mathcal{L}\_{logit}$ (Eq. 3) is a lower bound of the NCELoss $\mathcal{L}\_{NCE}$ (Eq. 2). I have never seen such result in (van de Oord et al., 2018) or anywhere.
- The loss Eq. 3 is indeed an upper bound of the negative mutual information (if the two sums in Eq. 4 is replaced by mean) although it has a different form from InfoNCE (van de Oord et al., 2018). I think the authors simply use the loss without really understanding what it is.
- I do not really understand how contrastive learning help disentangle the right factors (or directions) if the latent code z and the shift $\epsilon$ are sampled randomly for both query and positive samples. Could the authors elaborate more on this?

3) Clarity in presentation:
- It seems that the flipping of negative samples into positive samples in Eq. 7 plays an important role in the model’s performance but is not carefully analyzed in the paper. I am curious about the main cause of flipping. Is it due to the poor selection of positive and negative samples at the first step? How does the flipping rate change during learning? Could the authors show a curve of this? In Table 4, I see that negative flipping does not improve the performance if one-hot regularization is not enforced. Could the authors explain this?
- How MIG and DCI are computed for discovering-based methods that only use GAN? It seems that these discovering-based methods use no encoder which is required to compute MIG and DCI.

4) Choosing metrics:
- I think the authors should use more up-to-date disentanglement metrics (e.g. JEMMIG [1]) when evaluating their method. The problem with MIG and DCI is that MIG was shown to capture only modularity, DCI consists of 3 separate sub-metrics and each of them only captures one aspect of disentanglement [2]. Could the authors tell which sub-metric of DCI you are using for evaluation?

[1] Theory and Evaluation Metrics for Learning Disentangled Representations, Do & Tran, ICLR 2020

[2] Measuring Disentanglement - A Review of Metrics, Zaidi et al., 2020

---

> ### Author Response · Authors · 2021-11-21
> **Response to Reviewer sBQs (Part 1)**
>
> [Correctness of the method]
>
> - Absolute operator & $R_+$.
>
> Thank you for this suggestion. There is an absolute operator in Eq. 1, we agree with you and have followed your suggestion to modify $R$ to $R_+$ to make it more rigorous.
>
> - The sizes of the query key set and the positive key set are different.
>
> Our settings are different from previous contrastive methods where the inputs are already existing. In our setting, we need to inference the pre-trained generative models to sample positive samples. Sampling $B \times N$ positive samples (generating $B \times N$ samples using the generative model) will bring a high computational cost. Instead, we generate $N$ positive samples and use them as a shared one.  This is a specific setting in our context, and we have clarified it in our rebuttal version as we use a shared positive set for $B$ different queries to reduce the computational cost.
>
> - Formula of NCE in Eq. 2 is incorrect.
>
> Thanks for pointing it out and the sum in the numerator is our typo. We have fixed it in our rebuttal version.
>
> - BCELoss  (Eq. 3) is a lower bound of the NCELoss (Eq. 2) & About “We do understand contrastive learning”.
>
> We are sorry that our unrigorous statement and incorrect citation lead to your confusion, and we have modified our paper to clarify this. We believe that our rebuttal version can address this problem well.
>
> Here we would like to explain why we use BCELoss: the BCELoss is original from [b], an important related work of [a]. But InfoNCE is a more popular form of contrastive loss, and we first introduced it in our paper, which is more friendly for those readers unfamiliar with contrastive learning. We use BCELoss for the following reasons.
>
> (i) In recent years, several works have used this loss to achieve contrastive learning [c,d,f,g]. In addition, BCELoss is NCEloss in [b].
>
> (ii) We use BCEloss to save computational costs. Specifically,  in implementation, if we use InfoNCE loss, we need to call the function “torch.nn.CrossEntropyLoss” N times for each iteration (only 1 time “torch.nn.BCEWithLogitsLoss” for BCEloss).
>
> We would like to provide a further discussion on the relation between BCELoss and  NCEloss (details in Appendix F). The BCELoss can be regarded as the upper bound of NCELoss, roughly. However, it can not be mathematically proven in a rigorous way. We provide an analysis to illustrate this point. The main idea is the following: when minimizing BCELoss, it will approximate Eq. 1 (in Appendix F). And when minimizing InfoNCE loss, it will approximate Eq. 2 (in Appendix F). Eq.1 is the upper bound of Eq. 2. Most part of the analysis is motivated by [a,b].
>
> We have some specific designs to adapt contrastive learning into our context. We summarize these modifications and related reasons here. We will modify our paper to include this for better understanding.
>
> (i) In our setting, the definition of positive and negative is different. The positive pairs are the samples with the same direction. The negative pair are the samples with different directions. This definition is irrelevant to $z$ and $\epsilon$, and we randomly sample $z$ and $\epsilon$ to increase the diversity of samples.
>
> (ii) The samples are generated rather than from an existing dataset; thus, we can generate any specific number of queries,  positive keys, and negative keys. To cover more diversity, we choose multiple-query.  To reduce computation cost, we use a shared positive key set.

---

> ### Author Response · Authors · 2021-11-21
> **Response to Reviewer sBQs (Part 2)**
>
> [Clarity in presentation]
>
> - Main cause of flipping.
>
> We understand your concern, and we would like to provide more details here and we would like to add this to the main paper. The reason for flipping negative samples into positive samples is not because of the way we select positive and negative samples. It is because there indeed exist different directions with the same semantics (false negative). In order to make the contrastive loss actually work in our setting, we need to find these samples and flip them. The reason for existing of false negative samples (different directions with the same semantics)  in our setting are two-fold:  (i) since we do not know the actual number of semantic directions, we set the number of directions in Navigator to be large so that it is greater than the exact number of ground-truth semantic directions. (ii) It is the property of the generative model itself, and there are indeed some directions that have the same semantics.
>
> - Negative flipping does not improve the performance if one-hot regularization is not enforced.
>
> When negative flipping is applied but one-hot regularization is not enforced, one semantic will be encoded in multiple dimensions of the representation (as shown in Fig. 5). This kind of representation does not conform with disentangled representation defined by these metrics (each semantic is encoded in each dimension).
>
> - How to compute MIG and DCI for discovering-based methods.
>
> We should have indeed made this clearer. We described this in Sec. 4.1. “For these methods, we follow Khrulkov et al. (2021) to train an additional encoder to extract disentangled representation,” as stated in Sec. 4.1 in our paper. We will highlight it in the further revision.
>
> [Choosing metrics]
>
> - More up-to-date disentanglement metrics
>
> Thanks for the constructive suggestion. We have a comparison of our method and baselines using JEMMIG [e]. Due to the time limit, here we only present the results of (Baseline: $\beta$-TCVAE and DS, which are the best two baselines) on Shapes3D. For results on other datasets, we will add them in the final version. From the following table, we know that even on JEMMIG, our model still outperforms the baselines by a large margin.
>
> | Models | JEMMIG (↓) |
> | :-----:| :----: |
> | $\beta$-TCVAE |1.274±0.486 |
> | DS | 1.402±0.455 |
> | DisCo  | **0.796±0.427**  |
>
> [a]  Aäron van den Oord, Yazhe Li, and Oriol Vinyals. Representation learning with contrastive predictive coding.
>
> [b] Michael Gutmann and Aapo Hyvärinen. Noise-contrastive estimation: A new estimation principle for unnormalized statistical models. AISTATS, 2010.
>
> [c] Wu, Zhirong, et al. "Unsupervised feature learning via non-parametric instance discrimination." CVPR, 2018.
>
> [d] Le-Khac, Phuc H., Graham Healy, and Alan F. Smeaton. "Contrastive representation learning: A framework and review." IEEE Access, 2020.
>
> [e] Do & Tran, “Theory and Evaluation Metrics for Learning Disentangled Representations”, ICLR 2020.
>
> [f] Mnih, Andriy, and Yee Whye Teh. "A fast and simple algorithm for training neural probabilistic language models." ICML, 2012.
>
> [g] Mnih, Andriy, and Koray Kavukcuoglu. "Learning word embeddings efficiently with noise-contrastive estimation." NeurIPS, 2013.

---

> ### Author Response · Authors · 2021-11-22
> **Looking forward to hearing from you**
>
> Dear Reviewer sBQs,
>
> We want to send you a friendly reminder for the discussion, since the second stage of discussion will be soon concluded.
>
> We thank you again for your valuable comments, and we are happy to extend our response if you have any other concerns left.
>
> Thanks.

---

> ### Comment · Reviewer_sBQs · 2021-11-30
> **I raise my score to 6**
>
> Based on the authors' detailed responses, I raise my score to 6. However, I hope the authors will be careful in writing papers next time and should not make unjustified arguments without explicit mathematical proof.

---

### Official Review · Reviewer_j95X · 2021-11-02

**Correctness:** 4
**Technical Novelty And Significance:** 3
**Empirical Novelty And Significance:** 3
**Recommendation:** 6
**Confidence:** 4

**Main Review:**

Strengths:
- The approach does not require any specific training.
- There is no fixed generative model type: it can be applied to GANs, VAEs and Flow models.
- The method significantly outperforms previous models in terms of disentanglement metrics.
- The method is quite stable to random seeds.
- The authors provide a thorough ablation study, report the model accuracy with std due to random seeds, check the model sensitivity to the values of hyperparameter T.

Weaknesses:
- The approach requires many 'tricks' and parts to work: Navigator, Contrastor consisting of two weight sharing encoders, contrastive approach, hard negatives flipping. Each component requires its own set of hyperparameters. The overall performance gain is significant, and the necessity is partially covered in the Ablation study section. But I wonder if it is needed to have two encoders with shared weights or it is possible to have only one? Is it required to tune hyperparameters for every component?

**Summary Of The Paper:**

This paper presents a framework to model disentangled directions for pretrained models. Such an approach mitigates the problems with poor generation quality arising while training models with additional regularization terms to force disentanglement. The underlying idea is contrastive-based: similar image variations are caused by changing the same factors in contrast to the remaining image variations. The proposed framework is model-agnostic: it can be applied to GANs, VAEs and flow models.

**Summary Of The Review:**

Overall, the proposed method is outperforming previous approaches in terms of disentanglement scores. My main concern is the general complexity of the model.

UPD: I keep my score the same.

---

> ### Author Response · Authors · 2021-11-21
> **Response to Reviewer j95X**
>
> Thanks for reviewing our paper and appreciating our idea. We would like to address your concerns below.
>
> - The approach requires many 'tricks'.
>
> We understand this concern. We have probably not made our points clearly, and we would like to clarify them here. Before going into details, let us provide an overview of our framework first. In general, we are doing two things: (i) contrasting the image variation (Contrastor) (ii) finding direction in the space (Navigator). For implementation, we use a matrix as the Navigator. For the Contrastor, by using “two shared weight encoders,” we want to highlight how we get the “variation”. In the implementation, only one encoder is needed. We only have two trainable components, the matrix (Navigator) and encoder (Contrastor).
> In addition, the hard negative flipping is for better contrasting by removing the hard case (many different directions that carry the same semantic meaning).
>
> - Is it required to tune hyperparameters for every component?
>
> Thanks for pointing this out. We would like to highlight two points on the hyper-parameters.
>
> (i) Our hyper-parameters are shared across different models and datasets (as shown in Appendix A). We don’t need to tune the hyper-parameters for any specific models on those datasets.
>
> (ii) The hyperparameters that are sensitive for the performance have been studied in our paper (the ratio of positive and negative samples). The ratio is a crucial setting for contrastive learning methods [a].
>
> In addition, since our method does NOT need extra efforts to tune the hyperparameters (it works well on the default setting), it is easy to reproduce our results, and we also provide our code (https://bit.ly/3innjnr).
>
> [a] Khosla, Prannay, et al. "Supervised contrastive learning." NeurIPS. 2020.

---

### Decision · Program_Chairs · 2022-01-20

**Decision:**

Accept (Poster)

**Comment:**

The paper proposes a framework, named Disentaglement via Contrast (DisCo), to learn disentangled representations via contrastive learning on well-pretrained generative models. The method aims at simultaneously discovering semantically meaningful directions in pretrained generative models and training and encoder to extract them. The method uses contrastive learning where random samples perturbed along the discovered directions are regularised to be similar. The method is versatile and can be applied to various pretrained non-disentangled generative models including GAN, VAE, and Flow. Extensive experimental evaluation shows the benefits of the approach.

The authors provided a strong rebuttal addressing many of the concerns raised by the reviewers, including running new experiments (such as adding the JEMMIG metric to measure disentanglement as requested by Reviewer sBQs). This led to all reviewers recommending to accept the work.

The paper provides an exhaustive empirical evaluation testing several models and results are convincing. This was highlighted by all reviewers.

While the high level description of the method is clear, in practice the method is quite sophisticated requiring many heuristics (e.g. entropy-based domination loss or flipping hard negatives). This requires tuning several hyperparameters and complicates the message. This is mitigated by an ablation study presented by the authors highlighting the importance of each component. This was highlighted by Reviewer j95X and the AC agrees. The paper does provide implementation details, and reproducibility is not a concern.

Related to this point, Reviewer Go6R points out that the paper falls short in providing clear explanations on why the method is able to find meaningful semantic directions, and on where do the gains of the proposed model come from. While the paper could improve in this direction, the proposed empirical validation is convincing.

Overall the paper presents an interesting method that performs well in practice. All reviewers recommend accepting the work. The AC agrees with this recommendation.